# Identification of *slit3* as a locus affecting nicotine preference in zebrafish and human smoking behaviour

Judit García-González[1], Alistair J Brock[1], Matthew O Parker[2], Riva J Riley[1], David Joliffe[3], Ari Sudwarts[1], Muy-Teck Teh[4], Elisabeth M Busch-Nentwich[5,6], Derek L Stemple[5], Adrian R Martineau[3], Jaakko Kaprio[7,8], Teemu Palviainen[7], Valerie Kuan[9], Robert T Walton[3]*, Caroline H Brennan[1]*

[1]School of Biological and Chemical Sciences, Queen Mary, University of London, London, United Kingdom; [2]School of Pharmacy and Biomedical Science, University of Portsmouth, Portsmouth, United Kingdom; [3]Barts and The London School of Medicine and Dentistry, Blizard Institute, London, United Kingdom; [4]Centre for Immunobiology and Regenerative Medicine, Institute of Dentistry, Barts and The London School of Medicine and Dentistry, London, United Kingdom; [5]Wellcome Trust Sanger Institute, Cambridge, United Kingdom; [6]Cambridge Institute of Therapeutic Immunology & Infectious Disease (CITIID), Jeffrey Cheah Biomedical Centre, University of Cambridge, Cambridge, United Kingdom; [7]Institute for Molecular Medicine FIMM, HiLIFE, Helsinki, Finland; [8]Department of Public Health, Faculty of Medicine, University of Helsinki, Helsinki, Finland; [9]Institute of Cardiovascular Science, University College London, London, United Kingdom

*For correspondence:
r.walton@qmul.ac.uk (RTW);
c.h.brennan@qmul.ac.uk (CHB)

**Competing interests:** The authors declare that no competing interests exist.

**Abstract** To facilitate smoking genetics research we determined whether a screen of mutagenized zebrafish for nicotine preference could predict loci affecting smoking behaviour. From 30 screened $F_3$ sibling groups, where each was derived from an individual ethyl-nitrosurea mutagenized $F_0$ fish, two showed increased or decreased nicotine preference. Out of 25 inactivating mutations carried by the $F_3$ fish, one in the *slit3* gene segregated with increased nicotine preference in heterozygous individuals. Focussed SNP analysis of the human *SLIT3* locus in cohorts from UK (n=863) and Finland (n=1715) identified two variants associated with cigarette consumption and likelihood of cessation. Characterisation of *slit3* mutant larvae and adult fish revealed decreased sensitivity to the dopaminergic and serotonergic antagonist amisulpride, known to affect startle reflex that is correlated with addiction in humans, and increased *htr1aa* mRNA expression in mutant larvae. No effect on neuronal pathfinding was detected. These findings reveal a role for SLIT3 in development of pathways affecting responses to nicotine in zebrafish and smoking in humans.

## Introduction

Tobacco smoking is the leading preventable cause of death worldwide placing a heavy social and financial burden on society (***World Health Organization, 2017***; ***National Center for Chronic Disease Prevention and Health Promotion (US) Office on Smoking and Health, 2014***; ***Xu et al., 2015***). It is well established that aspects of smoking behaviour have a strong genetic component (***Munafò et al., 2004***; ***Batra et al., 2003***; ***Liu et al., 2019***; ***Erzurumluoglu et al., 2019***). However, identifying causal genetic factors and exploring the mechanisms by which they act is challenging in human studies: the field has been characterized by small effect sizes and lack of replication such that

there are remarkably few genes and loci that can be confidently linked to smoking. The strongest evidence for causal effects is for functional variants in *CHRNA5* (*Chen et al., 2015*) and *CYP2A6* (*Munafò et al., 2004*) affecting amount smoked and nicotine metabolism, respectively. Recent large studies have identified numerous new association loci, but their significance is yet to be biologically characterised (*Liu et al., 2019*; *Erzurumluoglu et al., 2019*).

As approaches to identify genetic risk are difficult in humans, research has been facilitated by studies in animal models, with a focus on genomic analysis of inbred and selectively-bred, naturally occurring genetic strains (*Crabbe, 2008*). This type of study produces quantitative trait loci (QTL) maps of multiple loci, each with a small impact on the phenotype. However, as with human studies, it is inherently difficult to identify relevant genes from QTL maps, as the overall phenotype cannot be predicted by individual genotypes. Mutagenesis studies in animal model systems can overcome these limitations: e.g. N-ethyl-N-nitrosourea (ENU) mutagenesis introduces thousands of point mutations into the genome with the potential to generate much stronger phenotypes than those occurring in a natural population thereby facilitating identification of causal mutations. Examination of phenotypic variation in ENU mutagenized model species could be applied to identify novel, naturally occurring variants influencing human addictive behaviour by identifying key genes and pathways affecting conserved behavioural phenotypes.

Drug-induced reinforcement of behaviour, that reflects the hedonic value of drugs of abuse including nicotine, is highly conserved in both mammalian and non-mammalian species (*Parker and Brennan, 2012*; *Engleman et al., 2016*; *Shipley et al., 2017*; *Berridge and Kringelbach, 2008*). Conditioned place preference (CPP), where drug exposure is paired with specific environmental cues, is commonly used as a measure of drug-induced reward or reinforcement (*Tzschentke, 1998*). ENU Mutagenesis screens for cocaine or amphetamine-induced CPP have been undertaken in zebrafish (*Darland and Dowling, 2001*; *Ninkovic et al., 2006*), however, despite successful isolation of lines with altered reinforcement responses to these drugs, the causal mutations have not been identified and the predictive validity of these screens for human behaviour has not been established. Larval locomotor response to nicotine has also been used to explore nicotine response genetics (*Petzold et al., 2009*) but the relevance of larval locomotion to addiction is somewhat less clear.

Here, we conducted a forward genetic screen of ENU-mutagenized zebrafish for nicotine-induced CPP. Zebrafish express homologues of all 16 members of the nicotinic acetylcholine receptor family present in mammals (*Zirger et al., 2003*; *Ackerman et al., 2009*; *Pedersen et al., 2019*) with similar binding characteristics (*Papke et al., 2012*; *Ponzoni et al., 2014*). However, as a result of a local gene duplication event in the ray fish lineage and a teleost genome tetraploidation event, zebrafish have duplicate copies of nicotinic receptor α2, α4, α7, α9, α10, β1, β3 and β5. In addition, zebrafish have additional receptors (α8 and α11) that have been lost in humans (*Pedersen et al., 2019*). Zebrafish show robust CPP to nicotine (*Ponzoni et al., 2014*; *Kily et al., 2008*; *Brock et al., 2017*; *Kedikian et al., 2013*). Nicotinic receptor partial agonists, that modulate striatal dopamine release in response to nicotine in mammalian systems, also inhibit nicotine-induced CPP in zebrafish (*Ponzoni et al., 2014*). Further, on prolonged exposure to nicotine or ethanol, adult zebrafish show conserved adaptive changes in gene expression and develop dependence-related behaviours, such as persistent drug seeking despite adverse stimuli or reinstatement of drug seeking following periods of abstinence (*Kily et al., 2008*; *Kedikian et al., 2013*). These data demonstrate the existence of a conserved nicotine-responsive reward pathway and support the suitability of zebrafish to examine the genetic and molecular mechanisms underlying behavioural responses to nicotine.

To evaluate the use of a behavioural CPP screen in zebrafish to predict loci affecting human smoking behaviour we initially assessed 1) the ability of varenicline and bupropion, pharmacological agents used to treat human nicotine addiction, to reduce zebrafish nicotine-induced place preference and 2) the heritability of nicotine responses in ENU-mutagenized fish. We then screened 30 $F_3$ families of ENU-mutagenized fish to identify families with increased/decreased CPP for nicotine. For two families with altered CPP response, the phenotype was confirmed following independent replication with a larger number of fish. Exome sequence information was used to generate a list of coding, loss of function candidate mutations affecting the phenotype. One family with a mutation co-segregating with increased nicotine CPP was selected for further study. Firstly, the effect of the identified gene on nicotine-induced CPP was confirmed using an independent line carrying a similar predicted loss of function mutation in the same gene. We then characterized the mutation using gene expression analysis, immunohistochemical analysis of neuronal pathways and behavioural

responses to acoustic startle; a response known to be modulated by serotonergic and dopaminergic signalling and, in humans, associated with vulnerability to addiction (*Loeber et al., 2007*; *Vrana et al., 2015*; *Kumari and Gray, 1999*). Finally, we used focused single nucleotide polymorphism (SNP) analysis of human cohorts to assess the predictive validity of findings in fish for human smoking behaviour.

In agreement with previous studies zebrafish showed a robust CPP to nicotine. Nicotine-induced CPP was abolished by varenicline and bupropion and found to be heritable in fish. The screening of ENU mutagenized families of zebrafish identified mutations in the *slit3* gene influencing sensitivity to rewarding effects of nicotine. *Slit3* mutant larvae and adult fish showed reduced behavioural sensitivity to amisulpride and larvae showed increased *ht1raa* receptor expression. No effect on neuronal pathfinding was detected. Analysis of the *SLIT3* locus in two independent human cohorts identified two genetic markers associated with level of cigarette consumption and likelihood of cessation. This proof of principle study demonstrates that screening of zebrafish is able to predict loci affecting complex human behavioural phenotypes and suggests a role for SLIT3 signalling in the development of dopaminergic and serotonergic pathways affecting behaviours associated with nicotine sensitivity.

## Results

### Nicotine CPP in zebrafish is inhibited by varenicline and buproprion

The hedonic value of drugs of abuse, that gives rise to reinforced behaviour, is commonly assessed using either self-administration protocols or CPP. The ability of compounds used as therapeutics in humans to prevent rodent nicotine self-administration is used to support the translational relevance of nicotine-self administration in that model (*Hall et al., 2015*; *O'Connor et al., 2010*; *Le Foll et al., 2012*). As our aim was to use nicotine-CPP to predict genes affecting smoking behaviour, we assessed the ability of the nicotine therapeutics varenicline and bupropion to inhibit nicotine induced CPP in zebrafish. As seen previously (*Kily et al., 2008*; *Kedikian et al., 2013*; *Brennan et al., 2011*), 10 µM nicotine induced a robust 15–20% change in place preference. Consistent with previous results (*Ponzoni et al., 2014*), pre-incubation in varenicline or bupropion dose-dependently inhibited the nicotine CPP response (*Figure 1*).

### Nicotine CPP is heritable in zebrafish

We selected a nicotine concentration predicted to induce a minimal detectable CPP in wild types (5 µM) (*Kily et al., 2008*; *Brock et al., 2017*), to enable us to detect both increased and decreased response to nicotine in mutants. To ensure that this strategy could detect genetic factors affecting response to nicotine, we assessed the heritability of the CPP response in ENU mutagenized fish using a selective breeding approach over three generations. *Figure 2A* shows our assessment strategy where fish showing the highest and lowest CPP response are selected for further breeding. In the first generation, the CPP change score phenotype was normally distributed (Shapiro-Wilks p=0.83) and there was a mean CPP change score of 0.11 to the drug paired side. CPP change scores ranged from -0.4 to 0.6.

An increasing difference in nicotine preference between offspring of fish from the upper vs lower extremes of the distribution (Shift of Cohen's $d$ = 0.89 in Second generation CPP to $d$ = 1.64 in Third generation CPP) indicates that nicotine CPP behaviour is heritable in zebrafish (*Figure 2B*), and that our CPP strategy is able to identify heritable differences in both extremes of the distribution. Phenotypes in the second and third generation screen may result from selecting for multiple co-segregating mutations that strengthened the phenotypes, or from selecting against other contrary mutations that weaken effects.

### Identification of *slit3* mutations affecting nicotine place preference in zebrafish

Out of 30 families screened, individuals from nine families were in the top 5% of the change in preference distribution, and individuals from five families in the bottom 5%. To identify candidate mutations affecting nicotine preference in fish, we focussed on families where all individuals included in the screen clustered at one or other extreme of the distribution curve. Two families (called AJBQM1 and AJBQM2 after the researcher who conducted the screen), which clustered at the top (AJBQM1)

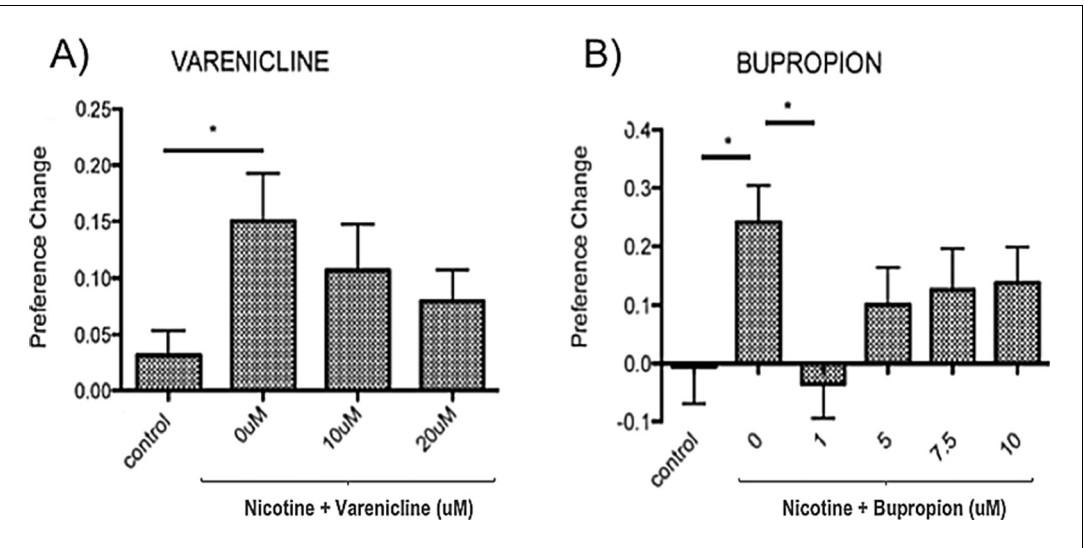

**Figure 1.** 10 µM nicotine induced place preference in zebrafish is sensitive to inhibition by therapeutics effective in humans. (**A**) Varenicline (nicotine partial agonist) and (**B**) Bupropion (norepinephrine and dopamine reuptake inhibitor with nicotine antagonist properties when metabolised). Bars represent mean and error bars represent + SEM. Asterisk (*) represents significance at p<0.05.

The online version of this article includes the following source data for figure 1:

**Source data 1.** Inhibition of nicotine CPP by varenicline and bupropion.

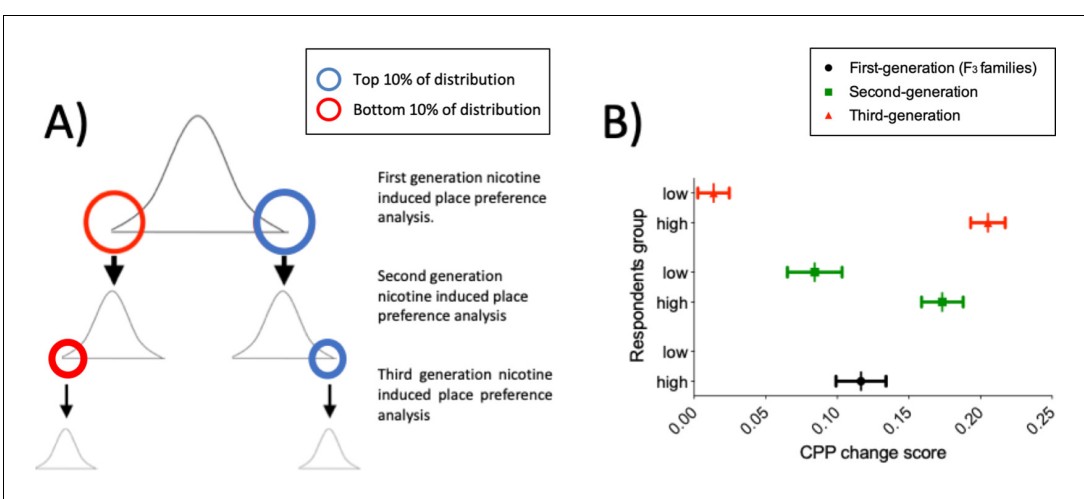

**Figure 2.** Nicotine CPP is heritable. (**A**) Breeding and selection to assess heritability of nicotine-induced place preference in ENU-mutagenized zebrafish. To test whether nicotine preference is heritable, fish in the upper (blue circle) and lower (red circle) 10% of the change in preference distribution curve were inbred and screened for CPP (Second generation CPP assay). A similar approach was used for the third generation CPP assay. (**B**) CPP for nicotine is heritable. Mean preference change is increasingly distinct for the second and third generation CPP analysis. Plot represents mean and ± SEM. First generation (corresponding to the $F_3$ families used for the screen) (n = 120): mean = 0.11; SD = 0.17. Second generation: Offspring of fish from *upper* 10% of the first generation screen (n = 92): mean = 0.17; SD = 0.14. Offspring of fish from *lower* 10% of the first generation screen (n = 64): mean = 0.08; SD = 0.15. Third generation. Offspring of fish from *upper* 10% of the second generation screen (n = 69): mean = 0.21; SD = 0.10. Offspring of fish from *lower* 10% of the second generation screen (n = 67): mean = 0.01; SD = 0.09.

The online version of this article includes the following source data for figure 2:

**Source data 1.** Nicotine CPP over three generations.

and bottom (AJBQM2) of the nicotine preference distribution, were selected for further study. We first assessed nicotine CPP in the remaining siblings not initially included in the screen. As shown in *Figure 3*, the phenotypes were conserved when remaining siblings were assessed. Exome sequencing of fish (*Kettleborough et al., 2013*) used to generate AJBQM1 and AJBQM2 identified 25 nonsense and essential splice site mutations. We genotyped fish at these 25 loci and determined the cosegregation with nicotine preference.

Of the 25 coding, predicted loss of function mutations in AJBQM1 and AJBQM2 (Listed in *Supplementary file 1A*), only *slit3$^{sa1569/+}$* (exon seven splice acceptor site disruption at amino acid position 176), segregated with nicotine preference (*Figure 4A* and *Supplementary file 1Ei*). None of the coding, predicted loss of function mutations in AJBQM2 segregated with nicotine preference and this line was not examined further (*Supplementary file 1Eii*).

To confirm that loss of *slit3* function was related to nicotine seeking behaviour we obtained an independent family of fish, *slit3$^{sa202}$*, carrying a G > T transversion producing a premature stop codon at amino acid position 163 in the Slit3 protein, from the Sanger Institute. Although not as marked as in AJBQM1 mutants (hereafter called *slit3$^{sa1569}$*), heterozygous *slit3$^{sa202}$* fish showed enhanced nicotine CPP (p=0.03) compared to wild type siblings (*Figure 4C and D*). The *slit3$^{sa1569}$* allele affects splicing and *slit3$^{sa202}$* introduces a premature stop codon. Both alleles reside before the second leucine rich repeat (LRR) domain in the encoded protein (*Figure 4B*), which is essential for interaction with ROBO receptor proteins (*Brose et al., 1999*).

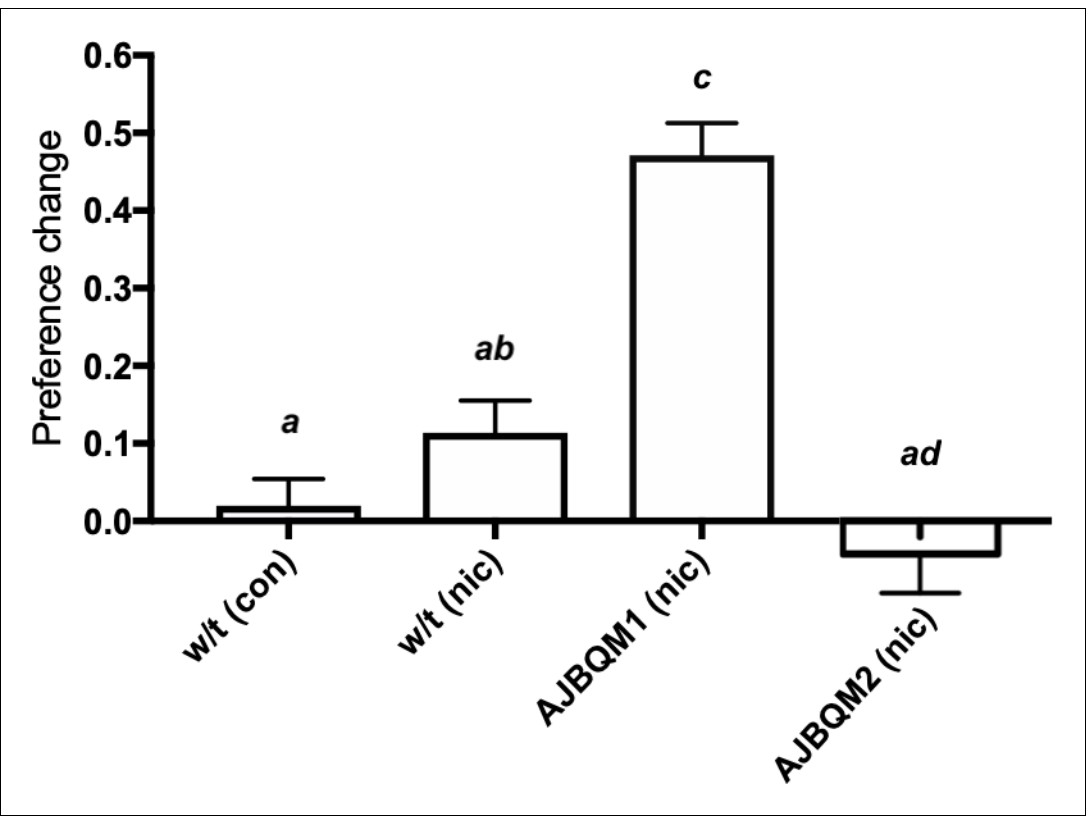

**Figure 3.** AJBQM1 and AJBQM2 families show increased and decreased nicotine place preference. AJBQM1 and AJBQM2 siblings, not included in the screen (n = 10 for AJBQM1; n = 14 for AJBQM2), AJBQM1 significantly differed from the parental strain, Tupfel longfin (TLF) wild type (w/t) saline control (n = 17) and wild type nicotine exposed fish (n = 7). AJBQM2 differed from wild type nicotine exposed fish but not wild type saline controls. Different superscript letters indicate significant difference between groups (p<0.05), same superscript letters indicate no significant differences between groups. Bars indicate Mean + SEM.
The online version of this article includes the following source data for figure 3:

**Source data 1.** Confirmation of nicotine CPP phenotypes for *Figure 3* and *Figure 4*.

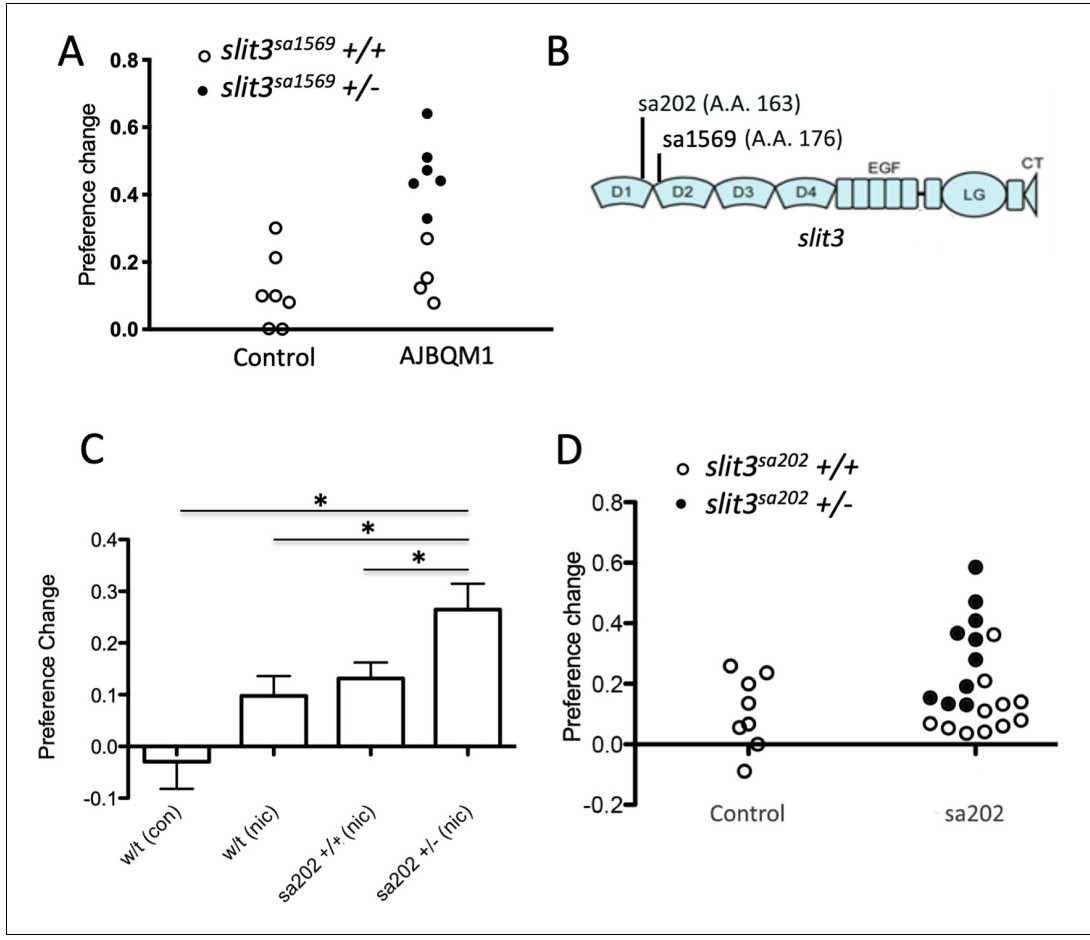

**Figure 4.** *slit3* mutations segregate with nicotine place preference. (**A**) Segregation of *slit3*<sup>sa1569</sup> mutation with nicotine seeking. CPP change scores for individual un-mutagenized TLF wild type fish (n = 7) and AJBQM1 fish (n = 10). Following CPP analysis, fish were genotyped for 25 loss of function mutations contained within the family. Black dots indicate *slit3*<sup>sa1569/+</sup> heterozygous mutant fish. White dots indicate *slit3*<sup>sa1569+/+</sup> fish. Heterozygosity for *slit3*<sup>sa1569</sup> segregates with increased nicotine seeking behaviour. (**B**) Position of ENU-induced mutations in zebrafish Slit3 protein. *slit3*<sup>sa1569</sup> (A > G transition) disrupts a splice site in intron seven affecting translation at amino acid 176. *slit3*<sup>sa202</sup> (G > T transversion) introduces a stop codon at amino acid 163. Both mutations truncate the protein before the leucine rich repeat domain 2 (D2), which interacts with membrane bound ROBO during SLIT-ROBO signalling. (**C**) Nicotine preference of *slit3*<sup>sa202</sup> line. *slit3*<sup>sa202/+</sup> fish (n = 18) show increased nicotine preference compared to wild type TLF controls (n = 8) (p=0.001) and wild type siblings *slit3*<sup>+/+</sup> (n = 14) (p<0.05). Bars indicate mean + SEM. (**D**) Segregation of *slit3*<sup>sa202</sup> allele with nicotine seeking. CPP change scores for individual un-mutagenised TLF wild type parent strain fish (n = 8) and *slit3*<sup>sa202</sup> fish (n = 21). Black dots indicate *slit3*<sup>sa202/+</sup> heterozygous mutant fish, white dots indicate *slit3*<sup>sa202+/+</sup> fish. Mutations in *slit3*<sup>sa202</sup> co-segregate with nicotine preference. Heterozygous *slit3*<sup>+/sa202</sup> present increased place preference compared to *slit3*<sup>sa202+/+</sup> siblings (n = 11).

## Characterisation of *Slit3*<sup>sa1569</sup> mutants

SLIT3 is a member of a family of proteins with established axon guidance properties and previously suggested to be involved in dopaminergic and serotonergic pathfinding (*Smidt and Burbach, 2007*). Therefore, we performed immunostaining in three-day-old zebrafish larvae and examined the number of cell bodies and axonal projections of serotonergic (5-HT) and catecholaminergic neurons in the brain using anti-5-HT and anti-tyrosine hydroxylase (TH) antibodies (*Figure 5*).

No differences between *slit3*<sup>sa1569</sup> mutant and wild type larvae were observed in the number of cells labelled by anti-5HT antibody in the raphe nucleus (Mean ± SEM: *slit3*<sup>+/+</sup>: 40.2 ± 2.6 vs. *slit3*<sup>sa1569/sa1569</sup>: 47.0 ± 4.7, p=0.23) rostral hypothalamus (*slit3*<sup>+/+</sup>: 15.4 ± 2.2 vs. *slit3*<sup>sa1569/sa1569</sup>: 19.4 ± 3.8, p=0.38) or inferior hypothalamic lobes (*slit3*<sup>+/+</sup>: 76.8 ± 13.5 vs. *slit3*<sup>sa1569/sa1569</sup>:

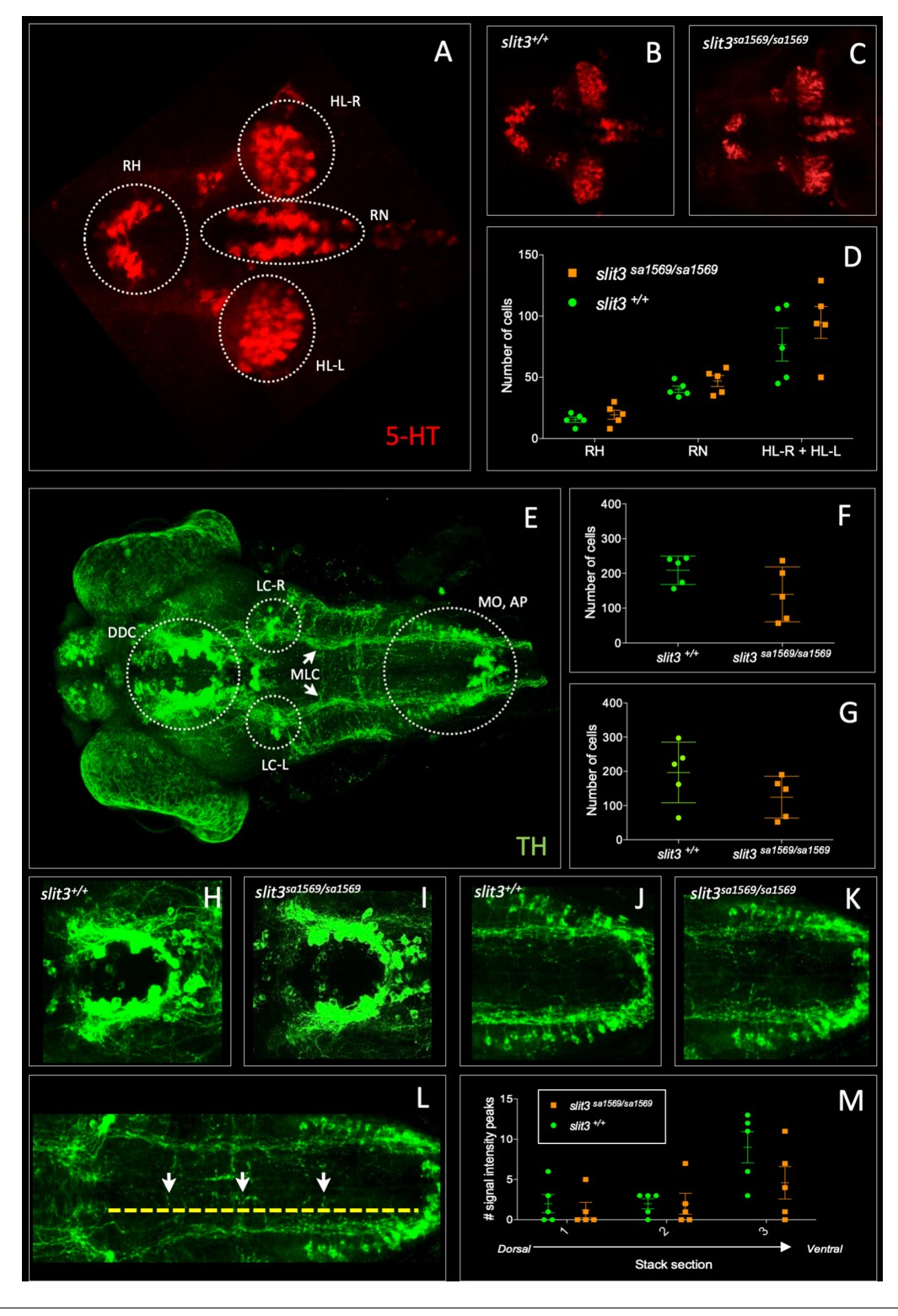

**Figure 5.** Fluorescent immunohistochemistry in three-day old wild type *slit3*<sup>sa1569+/+</sup> and homozygous mutant *slit3*<sup>sa1569/sa1569</sup>. (**A–D**) Anti-5-HT, (**E–M**) anti-TH: (**A**) 5-HT-labelled neurons in wild type zebrafish brain. Circles indicate regions used for quantification of cell number in rostral hypothalamus (RH), inferior hypothalamic lobes (HL-R, HL-L) and raphe nucleus (RN). (**B**) Anti-5-HT labelled cells in *slit3* wild type brain, (**C**) Anti-5-HT labelled cells in *slit3*<sup>sa1569</sup> homozygous mutant brain. (**D**) Quantification of anti-5HT labelled cell number in wild type and *slit3*

*Figure 5 continued on next page*

*Figure 5 continued*

mutant brains No significant differences were observed between wild type and *slit3* mutant larvae. (**E**) Unprocessed maximum intensity projection of anti-TH-labelled whole mounted wild type zebrafish brain. Circles indicate areas used for quantification, or in the case of LC-R and LC-L, landmarks used as reference to determine the extension of the medial longitudinal catecholaminergic tract (MLC) used when quantifying the number of anti-TH labelled projections to the midline (panels L, M). (**F**) Cell quantification for diencephalic dopaminergic cluster (DDC). No significant differences were observed between wild type and *slit3* mutant larvae. (**G**) Cell quantification for medulla oblongata interfascicular zone and vagal area, and area postrema (MO, AP). No significant differences were observed between wild type and *slit3* mutant larvae. (**H–K**) Anti-TH labelled wild types and *slit3^{sa1569}*. Zoomed-in visualization of diencephalic dopaminergic cluster (**H–I**) and medulla oblongata interfascicular zone and vagal area (**J–K**). (**L–M**) Quantification of catecholaminergic projections projecting to the midline. Examples of projections are indicated with yellow arrows. Projections were assessed from posterior to anterior using the locus coerulus and posterior extent of the raphe nucleus as landmarks (Panel L, yellow line) and from dorsal to ventral (Panel M, stacks 1–3). *Figure 5—figure supplement 1* shows individual planes. n = 5 samples per genotype group.

The online version of this article includes the following source data and figure supplement(s) for figure 5:

**Source data 1.** Quantification of anti-5HT and anti-TH labelled cell number and anti-TH labelled axon projections.
**Figure supplement 1.** Fluorescent immunohistochemistry in three-day old wild type and homozygous mutant *slit3^{sa1569}* labelled with tyrosine hydroxylase (**A**) and tubulin (**B–F**).

---

94.8 ± 12.9, p=0.36), nor in the number of cells labelled by anti-TH antibody in the diencephalic dopaminergic cluster (*slit3^{+/+}*: 209 ± 18 vs. *slit3^{sa1569/sa1569}*: 140 ± 35, p=0.12) and medulla oblongata interfascicular zone and vagal area (*slit3^{+/+}*: 196 ± 40 vs. *slit3^{sa1569/sa1569}*: 124 ± 27, p=0.17). Similarly, we observed no significant differences in the number of anti-TH labelled axon tracts projecting to the midline across the three planes examined (*slit3^{+/+}*: 2.6 ± 2.3 vs *slit3^{sa1569/sa1569}*: 4.3 ± 1, p=0.53) (*Figure 5* and *Figure 5—figure supplement 1*).

We also looked at the expression patterns using anti-acetylated tubulin antibody along the midline in the ventral forebrain, where *slit3* is known to be expressed (*Miyasaka et al., 2005*). However, no obvious differences were observed. Staining of *slbp^{ty77e/ty77e}* mutant larvae, known to have fewer neurons and axonal defects (*Petzold et al., 2009*) were used as positive control (*Figure 5—figure supplement 1*).

Although we did not observe differences between wild type and *slit3^{sa1569}* mutants, subtle effects on circuit formation may not have been detected by our antibody staining. To further characterise the *slit3* phenotype and explore potential functional differences in catecholamine circuitry, we examined the response and habituation to acoustic startle stimuli in wild type and mutant fish. Habituation to acoustic startle is known to involve catecholamine signalling and to be sensitive to dopaminergic/serotonergic antagonists such as amisulpride (*Quednow et al., 2006*) and, in humans, is associated with vulnerability to addiction (*Loeber et al., 2007*; *Vrana et al., 2015*; *Kumari and Gray, 1999*). Five-day-old larvae were subjected to 10 sound/vibration stimuli over a total of 20 s (2 s interval between each stimulus) in the presence of 0, 0.05 mg/L, 0.1 mg/L or 0.5 mg/L amisulpride in 0.05% dimethyl sulfoxide (DMSO). The distance travelled one second after each stimulus was recorded for each fish.

Response and habituation to the stimuli was quantified as the percentage of fish moving more than the mean plus two standard deviations of the mean baseline distance travelled per second before startle stimulus (mean + 2SD = 4.6 mm) in the first second after each stimulus. Using this criterion – and in line with the habituation response paradigm (*Rankin et al., 2009*) – a lower percentage of fish responded as the number of stimuli increased: 68% of wild type, non-drug-treated individuals responded to the first stimulus, 81% to the first and/or second stimulus, whereas 16% responded to the last stimulus (*Figure 6B*).

In drug-free conditions, there were no differences across *slit3^{sa1569}* genotype groups (*Figure 6C*). However, when larvae were treated with amisulpride, the habituation to startle responses across taps was different across genotypes. Amisulpride caused a biphasic dose dependent effect on stimulus response in wild types such that 0.05 mg/L caused an increase in responders across all 10 stimuli, and 0.5 mg/L caused a decrease (Effect of amisulpride dose p<0.001). A similar pattern was observed for heterozygous *slit3^{sa1569}*, but the effect of amisulpride was not significant (p=0.083).

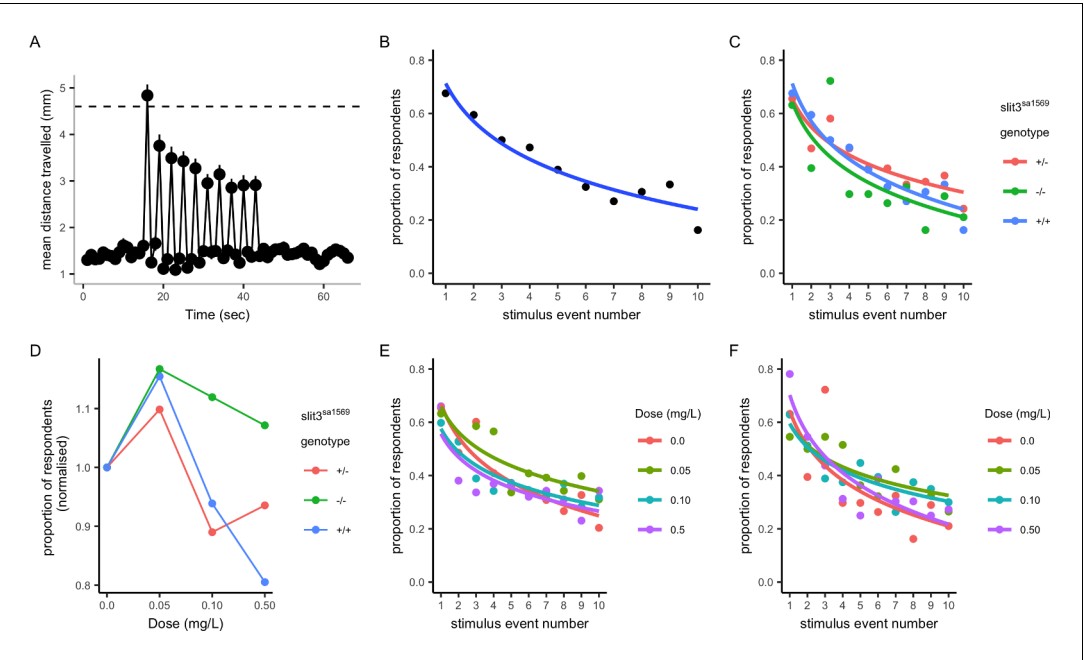

**Figure 6.** Habituation response in the presence and absence of amisulpride. (**A–B**) Response and habituation to 10 stimuli with two seconds interval between stimuli in wild type, drug free zebrafish. Mean distances travelled were measured in one second time bins. Line indicates 4.6 mm, which corresponds to mean basal distance moved per second plus 2 standard deviations of the mean and was used to define respondents. The percentage of fish responding to the stimuli decreases with stimulus/tap number (Main effects of tap number p<0.05) 68% respond to the first tap; 16% respond to the last tap. Respondents are defined as fish moving more than 4.6 mm. (**C**) Proportion of responders across the ten stimuli in drug free individuals from each genotype: there was no significant effect of genotype on response across taps (p=0.34) or responsiveness (p=0.35) in drug free fish. (**D**) Mean percentage of responders across the ten stimuli (± SEM). Data are stratified by *slit3^sa1569* genotype and amisulpride dose normalised to response in absence of drug. The effect of amisulpride on habituation varies by genotype. (**E, F**) Proportion of individuals responding in each amisulpride dose condition in wild type and homozygous mutant fish, respectively. The interaction between amisulpride dose and stimulus event number had a significant effect on the proportion of responsive individuals in wild type individuals (p<0.05) but not homozygous mutants (p=0.16).

The online version of this article includes the following source data, source code and figure supplement(s) for figure 6:

**Source code 1.** Zebrafish habituation to acoustic startle R code.
**Source data 1.** Response to acoustic startle in the presence and absence of amisulpride.
**Figure supplement 1.** Average distance moved before (*Figure 1A*) and during startle stimuli (*Figure 1B*) in wild type and *slit3^sa1569* mutant five-day-old zebrafish larvae.

Amisulpride dose had no significant effect on stimulus response in homozygous *slit3^sa1569*, that showed an increase in response to low doses but were less sensitive to inhibition at high doses (*Figure 6D–F*). The presence of a *slit3^sa1569* genotype by amisulpride dose interaction across taps was confirmed by a three-way interaction in the regression models (p=0.04). The interaction between dose and stimulus event number was a significant predictor of response in wild type larvae (p=0.044) and heterozygous larvae (p=0.02) but not in homozygous larvae (p=0.16).

There were no significant differences in locomotion before the first tap stimulus, in magnitude of the response to the first tap stimulus, nor in total distance moved across all tap stimuli across experimental groups (*Figure 6—figure supplement 1*) indicating that differences in startle behaviour were not confounded by differences in locomotion per se.

Adult *slit3 ^sa1569/ sa1569* mutant zebrafish showed a qualitatively different response to inhibition of CPP by amisulpride compared to wild type siblings, consistent with a persistent difference in sensitivity to this drug. The minimal CPP induced by 5 µM nicotine in wild type fish was prevented by

pre-exposure to 0.5 mg/L amisulpride. Nicotine-induced CPP in *slit3^sa1569* homozygous mutants was not affected (*Figure 7*).

As *slit3^sa1569* homozygous mutant fish showed altered sensitivity to nicotine and amisulpride, we examined whether expression of genes previously associated with nicotine dependence was dysregulated in *slit3^sa1569* mutant larvae using quantitative real-time PCR (qPCR). We included the nicotinic receptors and genes from the dopamine receptor family (*Huang et al., 2008a*; *Swan et al., 2005*; *Huang et al., 2008b*; *Sieminska et al., 2009*) -*drd1* (*Huang et al., 2008a*), *drd2* (*Swan et al., 2005*) and *drd3* (*Huang et al., 2008b*)-, and the dopamine transporter *dat* (*Sieminska et al., 2009*). Genes from the adreno-receptor families (*adra1* and *adra2*) were also included due to their links with nicotine addiction and use as putative targets for smoking treatment (*Forget et al., 2010*; *Kotagale et al., 2010*; *Swan et al., 2006*). Finally, expression of serotonin receptor genes was included, again due to their well-established links with nicotine addiction (*Kenny et al., 2001*; *Levin et al., 2008*; *de Bruin et al., 2013*; *Hauser et al., 2014*; *Bloch et al., 2010*).

For several genes (i.e. *drd3, chrnb3, htr4, adra2b*), up-regulation of gene expression in mutant larvae showed nominal significance (*Supplementary file 1H*). However, only *htr1aa* ([F(2,6)=44], p=0.0003) showed a significant difference across genotypes after correcting for multiple testing (*Figure 8*).

## Genetic variation at the *SLIT3* locus is associated with smoking behaviour in human samples

We next examined associations between 19 single nucleotide polymorphisms (SNPs) in the human *SLIT3* gene and smoking behaviour in two London cohorts. Two SNPs, rs12654448 and rs17734503

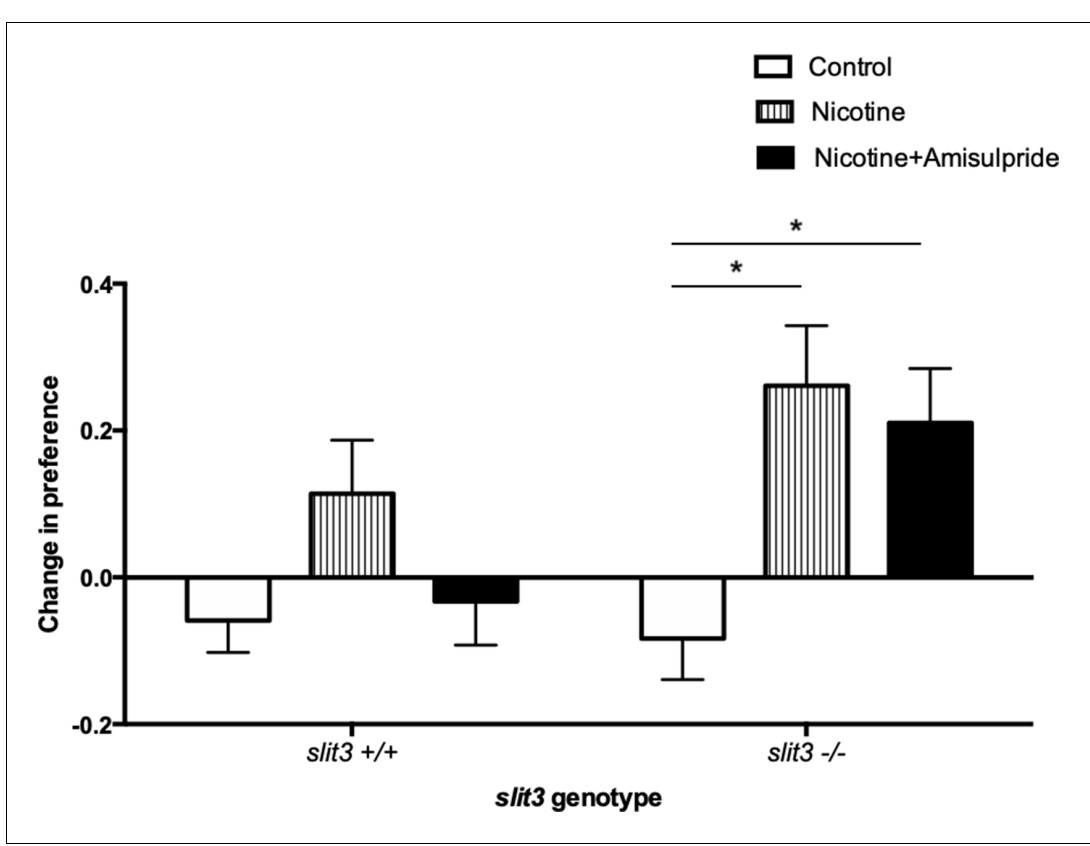

**Figure 7.** CPP induced by 5 µM nicotine is blocked by 0.5 mg/L dopamine/serotonin antagonist amisulpride in wild type *slit3^sa1569+/+* fish but not in *slit3^sa1569* homozygous mutants. Bars represent mean (+ SEM). (n = 11–14 fish per group). *Two-way ANOVA followed by post-hoc Tukey tests (p<0.05).

The online version of this article includes the following source data for figure 7:

**Source data 1.** Inhibition of nicotine CPP by amisulpride.

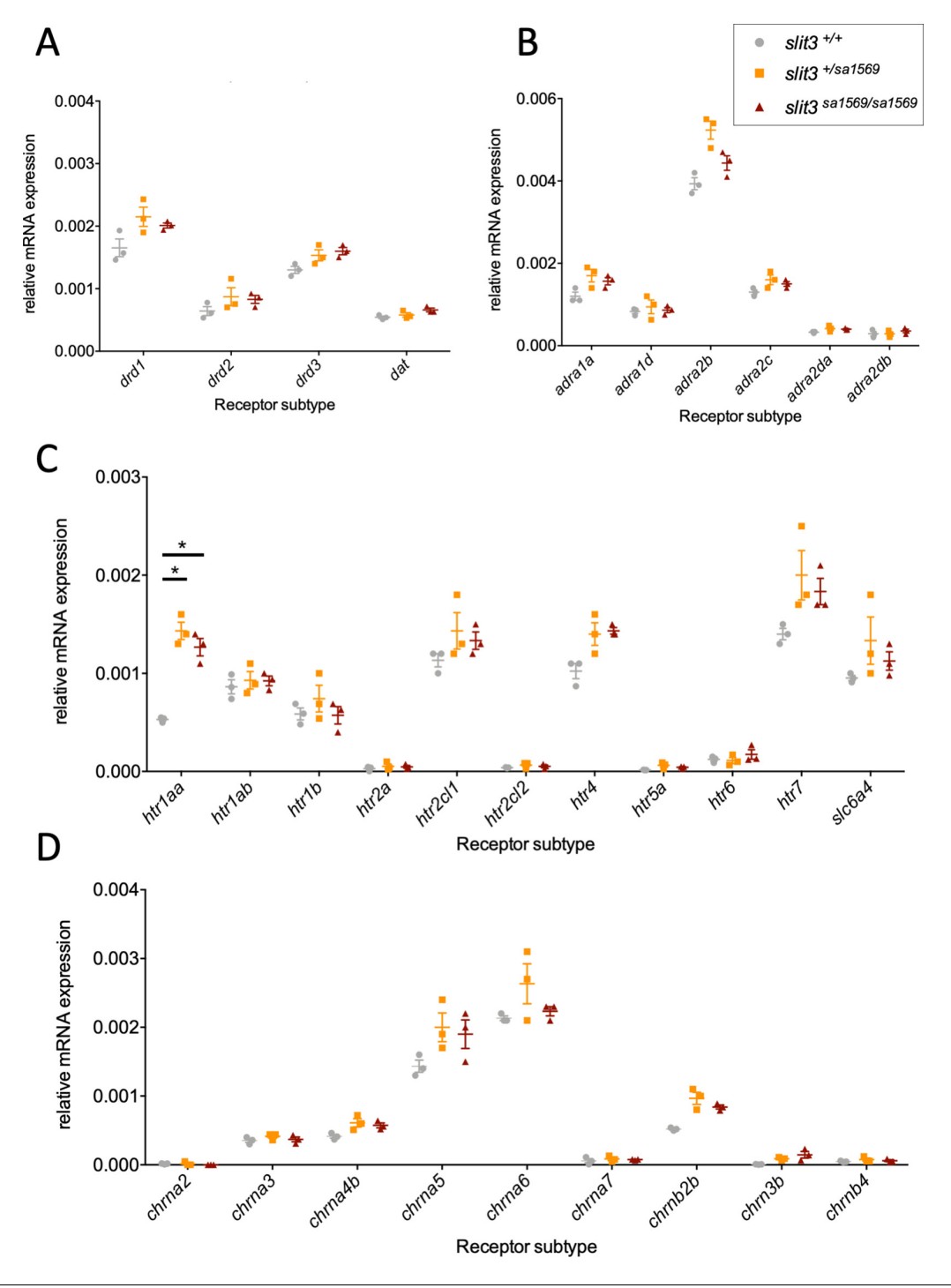

**Figure 8.** Quantitative real-time PCR analysis of five-day-old wild type *slit3sa1569+/+*, *slit3sa1569/+* heterozygous and *slit3sa1569/sa1569* homozygous mutant larvae. Quantitative PCR analysis of gene expression was performed for members of (**A**) dopaminergic signalling pathway, (**B**) adrenoreceptors, (**C**) serotonin signalling pathway and (**D**) nicotinic cholinergic receptors. (Total n=30, 3 samples per experimental group with n=10 embryos per sample). Only *htr1aa* ([$F_{(2,6)}$=44], p=0.0003) showed a significant difference across genotypes after correcting for multiple testing. *Two-way ANOVA followed by post-hoc Tukey test (p < 0.05).

The online version of this article includes the following source data for figure 8:

**Source data 1.** Gene expression data for *slit3* mutant and wild type zebrafish.

in high linkage disequilibrium (*Figure 9*) were associated with level of cigarette consumption (p=0.00125 and p=0.00227). We repeated the analysis on heavy smokers: rs12654448 (p=0.0003397) and rs17734503 (p=0.0008575) were again associated with cigarette consumption together with rs11742567 (p=0.004715). The SNP rs11742567 was associated with cigarette consumption in light smokers (<20 cigarettes per day, p=0.003909)) and with quitting. Associations are reported in *Table 1*. No other *SLIT3* polymorphisms were associated with smoking initiation, persistent smoking or cessation (*Supplementary file 1F and G*).

We subsequently investigated associations with more detailed smoking phenotypes in the Finnish twins cohort (*Kaprio, 2006*; *Table 2*). Associations were observed between rs17734503 and DSM-IV nicotine dependence symptoms (p=0.0322) and age at onset of weekly smoking (p=0.00116) and between rs12654448 and age at onset of weekly smoking (p=0.00105). Associations were seen elsewhere between *SLIT3* markers and Fagerström Test for Nicotine Dependence (FTND), cigarettes smoked each day, sensation felt after smoking first cigarette and time to first cigarette in the morning. In keeping with the London studies the minor allele was associated with a lower degree of dependence and decreased cigarette consumption.

The SNPs rs12654448 and rs17734503 are in non-coding domains, therefore it was not possible to predict loss or gain of function of SLIT3 from the SNP location. No evidence of affecting gene expression was found as per GTEx database (https://gtexportal.org/home/).

## Discussion

The aim of this study was to use forward genetic screening in zebrafish to identify loci affecting human smoking behaviour. Among 30 mutant zebrafish families screened for CPP, we identified one family showing increased nicotine preference compared to wild types. Out of the 13 pre-identified loss-of-function mutations in that family, only one in the *slit3* gene co-segregated with the behaviour. We confirmed the association between *slit3* loss of function and increased nicotine preference using

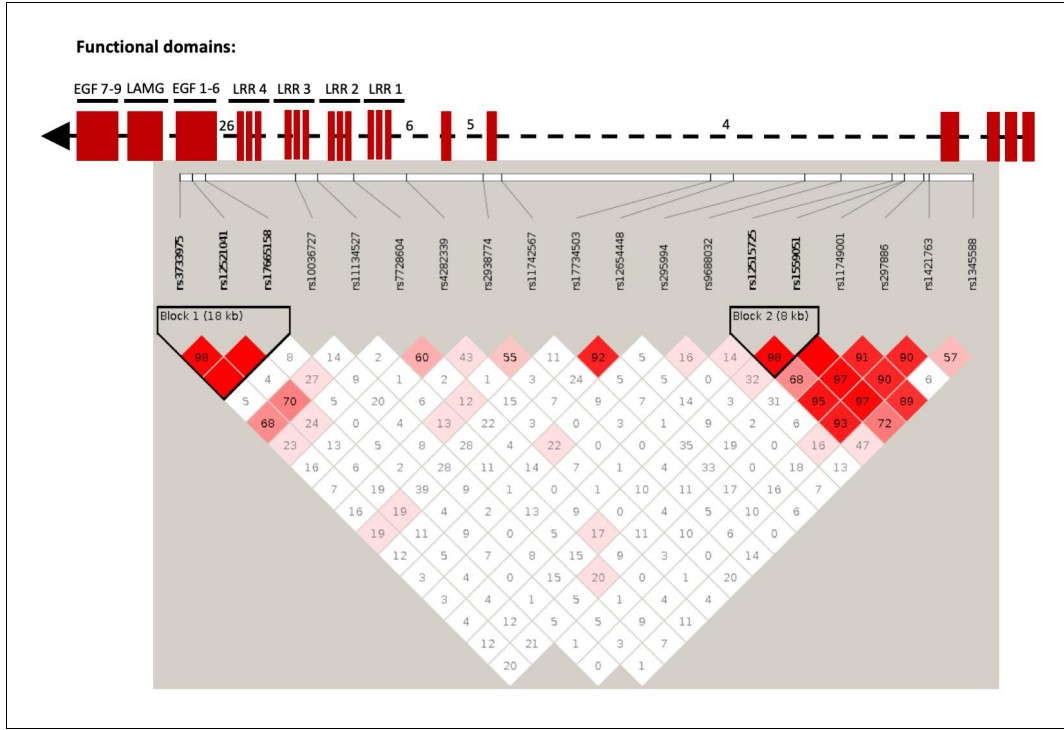

**Figure 9.** Linkage disequilibrium (LD) plot of *SLIT3* SNPs in human smoking association analysis. Numbers within each square indicate D' values (white: D'<1, LOD <2; blue: D'=1, LOD <2; pink: D'<1, LOD ≥2; and bright red: D'=1, LOD ≥2). Top part of the figure shows domain organization of the SLIT protein based on the UCSC Genome Browser (http://genome.ucsc.edu/) in relation to the SNP location. LRR: leucin-rich repeats. EGF: epidermal growth factor domains. LamG; Laminin G domain. Some intron numbers were added for reference.

**Table 1.** Associations of *SLIT3* SNPs with level of tobacco consumption for the London study groups (n = 863).

Regression coefficients, confidence intervals and p-values from linear regression of cigarettes smoked per day (CPD) on minor allele count for smokers from COPD, asthma and general cohorts, adjusted for age, sex and cohort. β coefficient represents effect of each additional minor allele. Benjamini-Hochberg cut-off at q-value 0.1 = 0.01053. Associations of *SLIT3* SNPs with tobacco consumption in a subset of heavy smokers (≥20 cigs/day). Adjusted for age, sex and cohort. (q-value 0.1 = 0.01579). Associations of *SLIT3* SNPs in a subset of light smokers (<20 cigs/day). Adjusted for age, sex and cohort (q-value 0.1 = 0.00526). Association analysis of *SLIT3* SNPs with smoking cessation. Logistic regression of current smokers *vs* ever smokers controlling for age, sex and cohort. Odds ratio >1 indicates minor allele increases odds of persistent smoking relative to major allele. SE: standard error, L95: lower limit of 95% confidence interval, U95: upper limit. For all panels, associations in bold remained significant after adjustment for multiple comparisons using a Benjamini-Hochberg procedure to control false discovery rate at 10%.

| SNP | Tobacco consumption | | | | Tobacco consumption - heavy smokers (≥20 cigs/day) | | | | Tobacco consumption - light smokers (<20 cigs/day) | | | | Smoking cessation | | | | |
|---|---|---|---|---|---|---|---|---|---|---|---|---|---|---|---|---|---|
| | P value | β | SE | 95% | P value | β | SE | 95% | P value | β | SE | 95% | OR | SE | L95 | U95 | P value |
| rs10036727 | 0.629 | −0.388 | 0.802 | (−1.960, 1.183) | 0.448 | −0.653 | 0.860 | (−2.337, 1.032) | 0.940 | −0.051 | 0.686 | (−1.396, 1.293) | 0.947 | 0.160 | 0.693 | 1.295 | 0.734 |
| rs11134527 | 0.218 | 1.014 | 0.822 | (−0.596, 2.625) | 0.327 | −0.867 | 0.883 | (−2.599, 0.864) | 0.261 | 0.795 | 0.705 | (−0.586, 2.176) | 0.665 | 0.165 | 0.482 | 0.918 | 0.013 |
| rs11742567 | 0.135 | −1.166 | 0.779 | (−2.691, 0.361) | **0.005** | **−2.346** | **0.825** | **(−3.962,−0.730)** | **0.004** | **1.888** | **0.644** | **(0.6258, 3.151)** | **1.586** | **0.163** | **1.153** | **2.183** | **0.005** |
| rs11749001 | 0.059 | 1.972 | 1.044 | (−0.074, 4.018) | 0.873 | 0.177 | 1.103 | (−1.985, 2.338) | 0.206 | 1.200 | 0.944 | (−0.651, 3.051) | 0.953 | 0.206 | 0.637 | 1.426 | 0.817 |
| rs12515725 | 0.592 | −0.406 | 0.756 | (−1.888, 1.076) | 0.488 | −0.565 | 0.813 | (−2.159, 1.029) | 0.278 | −0.688 | 0.631 | (−1.925, 0.550) | 1.028 | 0.151 | 0.765 | 1.381 | 0.855 |
| rs12521041 | 0.904 | −0.105 | 0.865 | (−1.801, 1.591) | 0.996 | −0.005 | 0.942 | (−1.851, 1.841) | 0.059 | −1.354 | 0.710 | (−2.746, 0.038) | 1.554 | 0.178 | 1.096 | 2.205 | 0.013 |
| rs12654448 | **0.001** | **−4.241** | **1.307** | **(−6.803, −1.680)** | **0.0003** | **−4.830** | **1.334** | **(−7.444, −2.216)** | 0.410 | −1.034 | 1.251 | (−3.486, 1.417) | 1.625 | 0.279 | 0.941 | 2.808 | 0.082 |
| rs1345588 | 0.240 | −1.268 | 1.078 | (−3.380, 0.845) | 0.253 | −1.334 | 1.164 | (−3.616, 0.948) | 0.869 | −0.150 | 0.907 | (−1.927, 1.627) | 1.417 | 0.222 | 0.918 | 2.189 | 0.116 |
| rs1421763 | 0.272 | −0.982 | 0.894 | (−2.735, 0.770) | 0.978 | −0.027 | 0.959 | (−1.908, 1.853) | 0.162 | −1.074 | 0.764 | (−2.571, 0.424) | 0.917 | 0.176 | 0.649 | 1.294 | 0.622 |
| rs1559051 | 0.961 | −0.040 | 0.819 | (−1.644, 1.564) | 0.458 | 0.656 | 0.882 | (−1.073, 2.384) | 0.507 | 0.455 | 0.685 | (−0.880, 1.797) | 0.919 | 0.163 | 0.668 | 1.265 | 0.606 |
| rs17665158 | 0.131 | 1.338 | 0.884 | (−0.394, 3.070) | 0.236 | 1.114 | 0.939 | (−0.727, 2.955) | 0.034 | 1.620 | 0.758 | (0.1354, 3.106) | 0.723 | 0.172 | 0.516 | 1.013 | 0.060 |
| rs17734503 | **0.002** | **−3.987** | **1.299** | **(−6.534, −1.441)** | **0.001** | **−4.458** | **1.325** | **(−7.055, −1.861)** | 0.410 | −1.034 | 1.251 | (−3.486, 1.417) | 1.616 | 0.275 | 0.942 | 2.773 | 0.081 |
| rs2938774 | 0.140 | 1.101 | 0.745 | (−0.359, 2.562) | 0.528 | 0.496 | 0.786 | (−1.044, 2.036) | 0.015 | −1.655 | 0.674 | (−2.976, −0.333) | 0.753 | 0.148 | 0.563 | 1.007 | 0.056 |
| rs295994 | 0.714 | 0.283 | 0.770 | (−1.227, 1.793) | 0.643 | 0.378 | 0.813 | (−1.215, 1.971) | 0.238 | −0.796 | 0.672 | (−2.114, 0.521) | 0.799 | 0.154 | 0.591 | 1.082 | 0.147 |
| rs297886 | 0.620 | 0.442 | 0.890 | (−1.303, 2.187) | 0.961 | −0.048 | 0.986 | (−1.979, 1.884) | 0.489 | 0.488 | 0.704 | (−0.891, 1.867) | 1.108 | 0.177 | 0.784 | 1.568 | 0.561 |
| rs3733975 | 0.909 | −0.099 | 0.860 | (−1.784, 1.587) | 0.982 | 0.022 | 0.934 | (−1.809, 1.852) | 0.059 | −1.354 | 0.710 | (−2.746, 0.038) | 1.488 | 0.176 | 1.054 | 2.101 | 0.024 |
| rs4282339 | 0.669 | −0.434 | 1.013 | (−2.419, 1.552) | 0.942 | −0.080 | 1.103 | (−2.241, 2.081) | 0.238 | −1.006 | 0.849 | (−2.670, 0.658) | 0.984 | 0.203 | 0.661 | 1.464 | 0.936 |
| rs7728604 | 0.701 | 0.286 | 0.744 | (−1.173, 1.745) | 0.321 | 0.827 | 0.832 | (−0.803, 2.457) | 0.654 | 0.262 | 0.583 | (−0.880, 1.404) | 0.935 | 0.149 | 0.698 | 1.253 | 0.653 |
| rs9688032 | 0.948 | −0.050 | 0.766 | (−1.551, 1.451) | 0.770 | 0.246 | 0.839 | (−1.398, 1.890) | 0.080 | −1.076 | 0.610 | (−2.272, 0.119) | 1.066 | 0.156 | 0.786 | 1.446 | 0.680 |

an independent zebrafish mutant line with a different loss of function mutation in *slit3*. Next, we established the relevance in humans by identifying two markers in *SLIT3* where the presence of the minor allele was associated with fewer cigarettes smoked each day and with smoking cessation. Studies in a separate twin cohort showed that these same alleles were associated with DSM-IV nicotine dependence symptoms and age at onset of weekly smoking. Taken together these findings suggest that zebrafish can be used to identify genes associated with smoking behaviour in humans and that variants in the *SLIT3* gene are linked in humans with a disruption of SLIT3 function that may affect propensity to develop tobacco dependence.

Consistent with previous findings in zebrafish and other species (*Ponzoni et al., 2014*), varenicline and bupropion inhibited nicotine induced place preference in zebrafish. The pattern of inhibition differed such that varenicline (partial agonist) showed increasing inhibition at increasing concentrations, whereas inhibition by bupropion (re-uptake inhibitor) was maximal at 1 μM, decreasing as concentration increased. The difference in response profile presumably reflects differences in the modes of action of these two compounds.

Our screening of ENU-mutagenized zebrafish families followed by sibling re-screen is a proof of principle study that indicates the relevance of zebrafish for human studies and emphasizes the

**Table 2.** Associations between detailed nicotine dependence phenotypes and *SLIT3* genotype in a Finnish twin cohort (n = 1715). Associations of *SLIT3* SNPs with DSM-IV nicotine diagnosis, symptoms, Fagerström scores (FTND), cigarettes smoked each day (CPD), age of onset of weekly smoking, sensation felt after smoking first cigarette and time to first cigarette in the morning. The three SNPs that were linked to smoking behaviour in the London cohorts are shown in bold.

| | SNP | DSM-IV ND diagnosis β | SE | P value | DSM-IV ND symptoms β | SE | P value | FTND (≥4) β | SE | P value | FTND score β | SE | P value |
|---|---|---|---|---|---|---|---|---|---|---|---|---|---|
| | rs12654448 | −0.0343 | 0.0262 | 0.190975 | −0.1839 | 0.0964 | 0.056728 | 0.0526 | 0.0287 | 0.066509 | 0.075 | 0.1365 | 0.58286 |
| | rs17734503 | −0.0354 | 0.0259 | 0.171821 | −0.2044 | 0.0954 | **0.032199** | 0.0474 | 0.0283 | 0.094383 | 0.0443 | 0.135 | 0.743052 |
| | rs11742567 | 0.0006 | 0.0163 | 0.97262 | −0.0359 | 0.0601 | 0.55086 | 0.0134 | 0.0179 | 0.45384 | 0.0449 | 0.0851 | 0.597682 |
| | rs17665158 | 0.0117 | 0.019 | 0.538639 | 0.1536 | 0.0696 | 0.027544 | 0.0178 | 0.0207 | 0.391096 | 0.0935 | 0.0988 | 0.344157 |
| | rs1345588 | −0.0031 | 0.0222 | 0.889847 | −0.0389 | 0.0817 | 0.634184 | 0.0578 | 0.0242 | 0.01708 | 0.1901 | 0.1157 | 0.100729 |
| | rs7728604 | −0.0049 | 0.0162 | 0.761485 | −0.0442 | 0.0597 | 0.459743 | 0.0004 | 0.0177 | 0.980706 | −0.0261 | 0.0846 | 0.757849 |
| | rs11134527 | 0.0296 | 0.0171 | 0.084576 | 0.0927 | 0.063 | 0.141369 | 0.0324 | 0.0187 | 0.083498 | 0.1376 | 0.0891 | 0.122807 |
| | rs10036727 | 0.0067 | 0.0165 | 0.68406 | 0.0207 | 0.0605 | 0.732266 | 0.0022 | 0.018 | 0.903865 | 0.046 | 0.0857 | 0.591583 |
| | rs1559051 | 0.0249 | 0.0193 | 0.198647 | 0.0492 | 0.0703 | 0.484353 | −0.0433 | 0.0208 | 0.037736 | −0.1011 | 0.0995 | 0.309836 |
| | rs12515725 | 0.0072 | 0.0159 | 0.6502 | 0.0277 | 0.0584 | 0.635739 | 0.0586 | 0.0172 | 0.000696 | 0.2482 | 0.0824 | 0.002637 |
| | rs2938774 | 0.0042 | 0.0173 | 0.80717 | 0.0096 | 0.0642 | 0.881054 | −0.0157 | 0.0191 | 0.41163 | −0.0173 | 0.0907 | 0.848978 |
| | rs295994 | −0.014 | 0.0171 | 0.410864 | −0.0144 | 0.0622 | 0.816397 | −0.016 | 0.0184 | 0.38542 | −0.0985 | 0.0879 | 0.262584 |
| | rs9688032 | −0.0174 | 0.0173 | 0.31299 | −0.0347 | 0.0636 | 0.585081 | 0.0234 | 0.0189 | 0.216144 | 0.0626 | 0.09 | 0.4869 |
| | rs11749001 | 0.0178 | 0.0235 | 0.448278 | −0.0062 | 0.0865 | 0.942516 | 0.0224 | 0.0257 | 0.383552 | 0.0067 | 0.1222 | 0.956097 |
| | rs4282339 | 0.0118 | 0.0201 | 0.557544 | 0.0526 | 0.0739 | 0.476216 | −0.0077 | 0.0219 | 0.724058 | 0.1623 | 0.1045 | 0.120641 |
| | rs297886 | −0.0216 | 0.0171 | 0.207835 | −0.045 | 0.0634 | 0.478517 | −0.0256 | 0.0188 | 0.173314 | −0.1354 | 0.0897 | 0.131469 |
| | rs1421763 | 0.0079 | 0.0187 | 0.671624 | 0.0178 | 0.0687 | 0.795522 | 0.0641 | 0.0203 | 0.001641 | 0.2582 | 0.0971 | 0.007892 |
| | rs3733975 | −0.013 | 0.0167 | 0.436903 | −0.0798 | 0.0613 | 0.192755 | −0.0384 | 0.0181 | 0.034371 | −0.2083 | 0.0866 | 0.016295 |
| | rs12521041 | −0.0098 | 0.0167 | 0.559173 | −0.0669 | 0.0613 | 0.275274 | −0.0365 | 0.0182 | 0.044962 | −0.1905 | 0.0868 | 0.028295 |

| SNP | CPD β | SE | P value | max CPD β | SE | P value | Age of onset of weekly smoking β | SE | P value | First time sensation β | SE | P value | FTND time to first cigarette β | SE | P value |
|---|---|---|---|---|---|---|---|---|---|---|---|---|---|---|---|
| **rs12654448** | -0.3509 | 0.5669 | 0.536029 | −1.0602 | 0.7743 | 0.171106 | 0.7826 | 0.2384 | 0.001051 | −0.0861 | 0.1423 | 0.545206 | 0.0047 | 0.0802 | 0.953291 |
| **rs17734503** | -0.479 | 0.5608 | 0.393086 | −1.2329 | 0.7657 | 0.107544 | 0.7689 | 0.2362 | 0.001156 | −0.1039 | 0.1406 | 0.460344 | 0.0188 | 0.0795 | 0.812682 |
| **rs11742567** | 0.0179 | 0.3532 | 0.959588 | −0.4621 | 0.4823 | 0.338159 | 0.0965 | 0.1493 | 0.518066 | −0.1003 | 0.0884 | 0.256427 | −0.0216 | 0.05 | 0.665265 |
| rs17665158 | 0.8135 | 0.4096 | 0.047191 | 1.5424 | 0.5587 | 0.005828 | 0.0562 | 0.1732 | 0.745385 | 0.2476 | 0.1027 | 0.01603 | −0.0884 | 0.058 | 0.12787 |
| rs1345588 | 0.294 | 0.4805 | 0.54066 | 0.3968 | 0.6562 | 0.54542 | 0.0989 | 0.2031 | 0.626431 | −0.0618 | 0.1204 | 0.607606 | −0.1303 | 0.068 | 0.055678 |
| rs7728604 | -0.0772 | 0.3511 | 0.825888 | 0.0691 | 0.4795 | 0.885363 | −0.0261 | 0.1486 | 0.860643 | −0.029 | 0.0875 | 0.740682 | −0.0193 | 0.0497 | 0.6978 |
| rs11134527 | 0.1831 | 0.3705 | 0.621187 | 0.7441 | 0.5057 | 0.141392 | −0.2347 | 0.1563 | 0.133517 | 0.1142 | 0.093 | 0.21965 | −0.1089 | 0.0523 | 0.037681 |
| rs10036727 | 0.1482 | 0.3557 | 0.67697 | 0.4246 | 0.4858 | 0.382161 | −0.0456 | 0.1507 | 0.762197 | 0.0289 | 0.0896 | 0.74711 | −0.0639 | 0.0504 | 0.205061 |
| rs1559051 | -0.4816 | 0.413 | 0.243693 | −0.4779 | 0.5641 | 0.397066 | 0.1437 | 0.175 | 0.411533 | −0.0289 | 0.1045 | 0.782381 | 0.0731 | 0.0586 | 0.212174 |
| rs12515725 | 0.5491 | 0.3429 | 0.10948 | 0.7708 | 0.4684 | 0.100032 | −0.1629 | 0.1452 | 0.26192 | −0.0368 | 0.0865 | 0.670165 | −0.1385 | 0.0485 | 0.00434 |
| rs2938774 | -0.2796 | 0.377 | 0.45835 | 0.1945 | 0.5149 | 0.70567 | −0.0575 | 0.1598 | 0.718862 | 0.043 | 0.0933 | 0.645221 | −0.0136 | 0.0533 | 0.797909 |
| rs295994 | -0.2793 | 0.3651 | 0.444276 | −0.2585 | 0.4988 | 0.604451 | 0.1543 | 0.1548 | 0.318928 | 0.0625 | 0.0926 | 0.499881 | 0.0527 | 0.0517 | 0.307869 |
| rs9688032 | 0.2452 | 0.3738 | 0.511921 | 0.4211 | 0.5107 | 0.409766 | −0.1142 | 0.1584 | 0.471283 | −0.2227 | 0.0937 | 0.01755 | −0.0517 | 0.0531 | 0.329867 |
| rs11749001 | -0.0301 | 0.5078 | 0.952789 | 0.0574 | 0.6939 | 0.934054 | −0.1994 | 0.2147 | 0.353064 | 0.1497 | 0.1274 | 0.240255 | 0.0075 | 0.0718 | 0.916823 |
| rs4282339 | 0.4086 | 0.434 | 0.346592 | 0.3083 | 0.593 | 0.603197 | 0.0952 | 0.1836 | 0.603958 | −0.0394 | 0.1084 | 0.716204 | −0.0524 | 0.0614 | 0.394221 |
| rs297886 | -0.1375 | 0.3727 | 0.712273 | −0.4782 | 0.5091 | 0.347622 | 0.1262 | 0.1575 | 0.423104 | 0.0519 | 0.0928 | 0.576255 | 0.0632 | 0.0527 | 0.230861 |
| rs1421763 | 0.5585 | 0.4037 | 0.166723 | 0.6702 | 0.5515 | 0.224417 | −0.1481 | 0.1706 | 0.385475 | −0.0799 | 0.1018 | 0.432497 | −0.1269 | 0.0571 | 0.026442 |

*Table 2 continued on next page*

*Table 2 continued*

| | | | DSM-IV ND diagnosis | | | DSM-IV ND symptoms | | | FTND (≥4) | | | FTND score | | | |
|---|---|---|---|---|---|---|---|---|---|---|---|---|---|---|---|
| rs3733975 | -0.7784 | 0.3597 | 0.030606 | −1.0555 | 0.4911 | 0.031758 | 0.0902 | 0.1521 | 0.553335 | −0.2932 | 0.0896 | 0.001085 | 0.1373 | 0.0509 | 0.007035 |
| rs12521041 | -0.7312 | 0.3602 | 0.042534 | −0.8864 | 0.492 | 0.071805 | 0.0522 | 0.1523 | 0.731943 | −0.3129 | 0.0897 | 0.000499 | 0.1257 | 0.051 | 0.01373 |

advantage of first using a screen with a low number of individuals per family to increase efficiency. The classic three generation forward genetic screen examines phenotypes in groups of 20 or more individuals from each family (*Lawson and Wolfe, 2011*). Logistical considerations make it difficult to apply such an approach to adult behavioural screens. Our approach increases efficiency by initially screening a small number of individuals from a large number of families and only selecting those families that occur at the extremes of the distribution for further analysis. Although in this study we were able to confirm phenotypes using a relatively small population of siblings, re-screening of a larger number would increase the power of the analysis and allow more subtle phenotypes to be identified.

One limitation of our forward genetic approach is that a large number of genes are duplicated in zebrafish due to the teleost tetraploidization; ENU-mutagenesis in one copy may not be sufficient to produce behavioural changes due to genetic compensation from the other copy of the same gene. *Slit3* is present as a single copy in the zebrafish genome which may have facilitated our ability to identify its role in responses to nicotine. However, arguably the most significant consequence of the teleost tetraploidization is the temporal and spatial specific expression of the gene duplicates. Spatial and/or temporal differences in gene expression patterns may offset concerns regarding compensation, and offer great potential to study region-specific functionality.

We identified a loss of function mutation in the zebrafish *slit3* gene associated with increased nicotine place preference and confirmed the phenotype in an independent line. SLIT molecules bind to ROBO receptors through a highly conserved leucine-rich repeat (LRR) domain (*Morlot et al., 2007*). In the AJBQM1 (*slit3*$^{sa1569}$) line the loss of function mutation causes a truncation at amino acid 176 and in the *slit3*$^{sa202}$ line at amino acid 163. These are immediately adjacent to the LRR2 domain responsible for SLIT3's functional interaction with ROBO proteins (*Morlot et al., 2007*) and would therefore be predicted to lead to formation of non-functional proteins. Initially identified as a family of axon guidance molecules, SLIT proteins are known to be expressed in a range of tissues and, by regulating cell polarity, to play major roles in many developmental process including cell migration, proliferation, adhesion, neuronal topographic map formation and dendritic spine remodelling (*Blockus and Chédotal, 2014*). In vitro SLIT proteins bind promiscuously to ROBO receptors suggesting that the proteins may co-operate in vivo in areas in which they overlap. However, their restricted spatial distributions, particularly of SLIT3 in the central nervous system (*Marillat et al., 2002*) suggest the individual proteins play subtly different roles in vivo.

Despite its neuronal expression, the most prominent phenotype seen in Slit3 deficient mice is postnatal diaphragmatic hernia (*Yuan et al., 2003*; *Liu et al., 2003*) with no obvious neuronal or axon pathfinding defects having been reported. Similarly, we did not detect any major differences in axon pathfinding nor in number of serotonergic and catecholaminergic cells in *slit3* mutant zebrafish larvae. As suggested previously (*Long et al., 2004*) it may be that overlap of expression with other slit molecules compensates for loss of slit3 in the brain preventing gross neuronal pathfinding defects. However, subtle differences in circuit formation and/or axon branching may have escaped our analysis.

As our antibody staining may not have detected subtle, functionally important differences in dopaminergic and/or serotonergic circuit formation we examined the impact of the dopaminergic and serotonergic antagonist amisulpride on the acoustic startle response, a behaviour associated with vulnerability to addiction and known to be sensitive to modulation by dopaminergic antagonists in zebrafish as well as mammals (*Quednow et al., 2006*; *Halberstadt and Geyer, 2009*; *Burgess and Granato, 2007*). Although the binding affinity of amisulpride in zebrafish has not been examined, reported behavioural effects (*Tran et al., 2015*) are consistent with binding characteristics and behaviours seen in mammalian species. In mammals, amisulpride binds to D2/3 pre-and post-synaptic receptors with greater affinity at presynaptic than post-synaptic receptors

(*Schoemaker et al., 1997*; *Perrault et al., 1997*). Presynaptic D2 receptors in mammals act as autoreceptors and inhibit the synthesis and subsequent release of dopamine. D2/3 postsynaptic receptors are coupled to Gi/o G proteins mediating inhibitory neurotransmission (*Beaulieu and Gainetdinov, 2011*). Treatment of rodents with D2/3 receptor antagonists leads to biphasic effects on locomotion such that low doses inhibit locomotion via pre-synaptic D2/3 autoreceptors, and high doses increase locomotion via post-synaptic D2/3 receptors (*Millan et al., 2004*). Treatment of adult zebrafish with amisulpride has a similar biphasic dose-dependent effect on locomotion such that low doses inhibit locomotion and high doses increase locomotion (*Tran et al., 2015*). These findings suggest that amisulpride has similar binding affinities at D2/3 receptors in fish as in rodents and imply the existence of pre-and post- synaptic D2/3 receptors with similar functional properties.

Our finding that high concentrations of amisulpride increased habituation to acoustic startle in wild type fish is in agreement with the effect of amisulpride in humans (*Quednow et al., 2006*). A biphasic dose-dependent effect of amisulpride on habituation in wild type larvae suggests the involvement of both pre- and post- synaptic dopamine receptors. Inhibition of presynaptic receptors at low dose leading to increased responsiveness (reduced habituation), and inhibition of postsynaptic receptors at high doses causing reduced responsiveness (increased habituation). In contrast to results in wild type fish, $slit3^{sa1569}$ mutant larvae showed decreased habituation in the presence of both high and low dose amisulpride, suggesting a reduction of sensitivity to amisulpride at post-synaptic sites, possibly related to the marginal increase in dopamine D3 receptors in *slit3* mutants. Differential sensitivity to amisulpride was also seen at adult stages, where amisulpride inhibited nicotine-induced CPP in adult wild type fish but not in $slit3^{sa1569}$ mutants. These findings are consistent with a disrupted dopaminergic system, and/or disrupted interactions between dopaminergic and serotonergic systems caused by *slit3* loss of function.

Although our results are consistent with differences in dopaminergic signalling, differential sensitivity to actions of amisulpride at serotonergic receptors cannot be ruled out: amisulpride also acts as an antagonist at Htr7 and Htr2b receptors with affinities approximately four times lower than at dopaminergic receptors. In mice, acoustic startle is sensitive to inhibition at Htr2b sites and genetic ablation of the *Htr2b* gene induced a reduction in startle amplitude and a deficit in prepulse inhibition of the startle reflex in loss of function mice (*Pitychoutis et al., 2015*). Loss of function of *Htr7* has no effect on acoustic startle or pre-pulse inhibition of acoustic startle in mice (*Semenova et al., 2008*).

Although gene expression analyses revealed subtle up-regulation in several receptors - including *drd3, chrnb3, adra2b* and *htr4* - in *slit3* mutants, significant difference was only seen for the *htr1aa* receptor subtype. Zebrafish possess two homologues of the *Htr1a* gene, *htr1aa* and *htr1ab*, with overlapping expression domains (*Norton et al., 2008*). The observation of increased *htr1aa* expression in *slit3* mutants is of interest: Serotonergic signalling has been previously linked to drug reward processes including nicotine use and dependence (*Fletcher et al., 2008*; *Olausson et al., 2002*). Manipulations which decrease brain serotonin neurotransmission (e.g., a neurotoxic serotonin depletion or a lasting serotonin synthesis inhibition) elevate self-administration of several different drugs including nicotine in rats (*Olausson et al., 2002*; *Roberts et al., 1994*; *LeMarquand et al., 1994*). Compounds that facilitate serotonin neurotransmission, such as selective serotonin reuptake inhibitors, decrease nicotine intake (*Opitz and Weischer, 1988*) however, the HTR1A specific antagonist WAY100635 has also been reported to block nicotine enhancement of cocaine and methamphetamine self-administration in adolescent rats (*Dao et al., 2011*). Nicotine increases serotonin release in the striatum, hippocampus, cortex, dorsal raphe ́nucleus (DRN), spinal cord and hypothalamus (*Seth et al., 2002*). The effects in the cortex, hippocampus, and DRN involve stimulation of *Htr1a* receptors, and in the striatum, *Htr3* receptors. In the DRN, *Htr1a* receptors play a role in mediating the anxiolytic effects of nicotine. In contrast, in the dorsal hippocampus and lateral septum, these same receptors mediate its anxiogenic effects. Further, pharmacological studies in rodents have shown that the *Htr1a* receptor antagonists WAY100635 and LY426965 alleviate the anxiety-related behavioural responses induced by nicotine withdrawal (*Rasmussen et al., 1997*; *Rasmussen et al., 2000*; *Harrison et al., 2001*). Although it is possible that an anxiolytic effect of nicotine contributed to the increased nicotine-induced place preference, preliminary assessment of anxiety-like responses (tank diving) in *slit3* mutants, where mutants show decreased anxiety-like behaviour (n.s), argue against this.

It is perhaps of particular interest that it is a homologue of the HTR1A receptor that is up-regulated in *slit3* mutants. HTR1A is the major inhibitory serotonergic receptor in mammalian systems. In mammals, it is present as autoreceptors on cell bodies and dendrites in the raphe nucleus and as post-synaptic heteroreceptors in brain regions implicated in mood and anxiety such as prefrontal cortex, hippocampus and amygdala. Projections from the raphe release serotonin throughout the entire forebrain. and brainstem and modulate a range of activities with additional raphe nuclei also providing innervation to the midbrain (see *Garcia-Garcia et al., 2014* for review). Interaction between HTR1A receptor and dopaminergic signalling are well established. Systemic administration of HTR1A receptor agonists leads to increased dopamine transmission in the nigrostrial pathway, ventral tegmental area and frontal cortex (*Garcia-Garcia et al., 2014*; *Bantick et al., 2001*). Although the mechanism underlying this increased dopamine transmission is not clear in all areas, the likely mechanism is by action on autoreceptors in the raphe so inhibiting serotonergic projections and disinhibiting dopaminergic transmission (*Bantick et al., 2001*). Within the nucleus accumbens, a key area in drug reward, HTR1A agonists have little effect on dopamine release under baseline conditions, but inhibit amphetamine induced release (*Ichikawa et al., 1995*). Further evidence for interactions between dopaminergic D2 receptor systems and HTR1A receptor signalling come from studies of atypical neuroleptics for the treatment of schizophrenia. Considerable evidence suggests that the balance between the properties of D2 receptors and HTR1A receptors influences the profile of action of these drugs in preclinical models (*Newman-Tancredi, 2010*; *Newman-Tancredi and Kleven, 2011*). Whilst synergistic effects of atypical neuroleptics may enhance dopamine release via inhibition of D2 autoreceptors and secondarily via disinhibition of projections from the raphe, it has been suggested (*Łukasiewicz et al., 2016*) that association of D2 receptors and HTR1A in functional heterodimers that exhibit properties distinct to either G protein coupled receptor may also be involved. Such an interaction is seen between D2 receptors and adenosine A2a receptors both in vitro and in vivo and has been reported for D2 receptors and HTR1A in vitro (*Łukasiewicz et al., 2016*). Whilst speculative, it is also potentially of relevance that HTR1A protein expression is upregulated in the brains of schizophrenics (*Carrard et al., 2011*) and variants in both *HTR1A* and *SLIT3* are associated with psychiatric disorders (*Shi et al., 2004*; *Glessner et al., 2010*; *Cukier et al., 2014*), known to involve dopaminergic and serotoninergic pathways.

The mechanism by which loss of function in *slit3* leads to increased *htr1aa* expression and disrupted dopaminergic signalling in zebrafish mutants is yet to be established. However, *HTR1A* is upregulated in conditions of reduced serotonergic signalling in other systems (*Garcia-Garcia et al., 2014*; *Chen et al., 1998*). Loss of serotonergic signalling from the DRN affects dopaminergic axonal outgrowth to the rat medial prefrontal cortex at developmental stages (*Benes et al., 2000*) such that lesioning of the DRN at neonatal stages results in significant increase in dopaminergic fibres. These effects are stage specific, raising the possibility that, although we did not detect any differences in catecholamine axon projections at three days post fertilisation, more detailed analysis at later stages of development would reveal significant differences. The observation that *slit3* is strongly expressed in the posterior raphe nucleus overlapping with *htr1aa* expression supports interaction between these two systems. Although detailed co-expression studies have not been performed, each of the other genes found to have marginal changes in expression also show overlapping expression with *slit3* (*Miyasaka et al., 2005*; *Norton et al., 2008*; *Boehmler et al., 2004*; *Ackerman and Boyd, 2016*).

Thus, our findings of an increase in *htr1aa* expression, altered sensitivity to amisulpride and altered nicotine CPP support a role for *slit3* signalling in the formation of dopaminergic and serotonergic pathways involved in responses to nicotine.

There are limitations to our findings: we used zebrafish of various ages in different experiments. While this confirms that loss-of-function in *slit3* alters behaviour from early life to adulthood, suggesting a developmental role, mechanisms underlying the two behavioural phenotypes may differ. We used whole embryos for the qPCR study so changes in expression in non-neuronal tissue may contribute to the observed differences, further, expression of genes in one tissues may mask changes of expression in another. In addition, we only examined a limited number of receptors and transporters for key neurotransmitter pathways. Important differences in other transmitter pathways and neurotransmitter metabolism may have been missed. We confirmed the translational effects of *SLIT3* gene variants in a human study, and the association was validated in a second, independent cohort. The sample sizes used in the human studies are small in comparison with those used in

human discovery studies, however we used the two human studies to validate the findings in fish and analysed only a small set of SNPs focussed on just one gene which minimises the chance of type one error. Analyses were also corrected for multiple comparisons. The Finnish cohort was not used as a formal replication of the findings in the London cohort but instead to explore the effects of genetic variation on richer and more informative smoking phenotypes.

Associations between *SLIT3* and aspects of smoking phenotype have also been found in previous GWAS (https://atlas.ctglab.nl) (*Watanabe et al., 2019*). However, larger studies would be necessary to obtain greater precision on estimates of the effect size. Further studies are also required to determine the effects of genetic variation in *SLIT3* on anatomical pathways in the human brain and their functioning with view to identifying people who are at high risk of developing dependence. This could be achieved by using imaging techniques to study brain activation in response to smoking related cues in smokers who have the *SLIT3* polymorphisms linked to smoking (particularly rs12654448).

To our knowledge, this is the first report of a forward behavioural genetic screen in adult zebrafish successfully predicting a novel human coding genetic region involved in a complex human behavioural trait. Taken together, these results provide evidence for a role for *SLIT3* in regulating smoking behaviour in humans and confirm adult zebrafish as a translationally relevant animal model for exploration of addiction-related behaviours. Further work analysing the cellular processes affected as a result of the *slit3* mutation may provide useful targets when designing tailored treatments to aid smoking cessation.

# Materials and methods

**Key resources table**

| Reagent type (species) or resource | Designation | Source or reference | Identifiers | Additional information |
|---|---|---|---|---|
| Genetic reagent (*Danio rerio*) | *slit3* | Sanger Institute | sa202 | Generated by ENU mutagenesis. Soon to be available from ZIRC zebrafish resource centre |
| Genetic reagent (*Danio rerio*) | *slit3* | Sanger Institute | sa1569 | Generated by ENU mutagenesis. Soon to be available from ZIRC zebrafish resource centre |
| Strain, strain background (*Danio rerio*) | Tupfel | Sanger Institute | Tubingen longfin | Wild type strain, now available from the ZIRC zebrafish resource centre |
| Antibody | anti-5-HT (Rabbit polyclonal) | Sigma | Cat#S5545, | IHC (1/200) |
| Antibody | anti-tyrosine hydroxylase (mouse monoclonal) | Abcam | Cat# AB152 | IHC (1/1000)) |
| Chemical compound | Amisulpride | Tocris | C2132 | 0.05–0.5 mg/L |
| Chemical compound | varenicline | sigma | PZ0004 | 10-20micromolar |
| Chemical compound | bupropion | Sigma | B1277 | 1–10 micromolar |
| Chemical compound | Nicotine hemisulphate | Sigma | N1019 | 5–10 micromolar |

## Animals

All in vivo experimental work was carried out following consultation of the ARRIVE guidelines (NC3Rs, UK). Required sample size was estimated following pilot studies to determine effect sizes,

and power calculations (beta = 0.8, alpha = 0.05). All animals were selected at random from groups of conspecifics for testing.

## Generation of F3 families of ENU-mutagenised fish

Wild type and ENU-mutagenized Tupfel longfin (TLF) fish were obtained from the Sanger Institute, as part of the Zebrafish Mutation Project which aimed to create a knockout allele in every zebrafish protein-coding gene (https://www.sanger.ac.uk/resources/zebrafish/zmp/). At the Sanger, ENU-mutagenized TLF $F_0$ males were outcrossed to create a population of $F_1$ fish heterozygous for ENU-induced mutations. Due to the high ENU mutation rate (1/300 kb in $F_1$ fish) and homologous recombination when $F_1$ gametes are generated, all $F_2$ were heterozygous for multiple mutations. $F_2$ families, each generated from a separate $F_1$ fish, were imported from the Sanger Institute to Queen Mary University of London (QMUL).

At QMUL a single male and female fish from each $F_2$ family were inbred to generate 30 $F_3$ families that would be 25% wild type, 50% heterozygous and 25% homozygous mutant for any single mutation, assuming Mendelian genetics. Based on exome sequencing data from the $F_1$ generation performed at the Sanger, each $F_3$ family contained 10–20 known predicted loss of function exonic mutations, approximately 100 non-synonymous coding mutations and approximately 1500 unknown mutations in non-coding domains across the entire genome (*Kettleborough et al., 2013*). Breeding scheme is detailed in *Figure 10*.

## Fish maintenance

Fish were housed in a recirculating system (Tecniplast, UK) on a 14 hr:10 hr light:dark cycle (08:30–22:30). The housing and testing rooms were maintained at ~25–28°C. Fish were maintained in aquarium-treated water and fed three times daily with live artemia (twice daily) and flake food (once). All procedures were carried out under license in accordance with the Animals (Scientific Procedures) Act, 1986 and under guidance from the local animal welfare and ethical review board at QMUL.

## Conditioned place preference (CPP)

All fish were age and weight matched for all behavioural analysis and were approximately 5 months old, weighing 0.2–0.25 g at the start of testing. Following habituation and determination of basal preference, animals were conditioned to 5 µM nicotine (Sigma, Gillingham, UK Catalogue number: N1019) over three consecutive days and assessed for a change in place preference the following day. 5 µM nicotine was used because it was predicted to induce a minimum detectable change in place preference based on results of previous studies (*Kily et al., 2008*; *Kedikian et al., 2013*). This minimal effective dose was used to avoid possible ceiling effects if using a higher concentration. CPP was assessed as described previously (*Kily et al., 2008*; *Brock et al., 2017*; *Parker et al., 2016*): The testing apparatus was an opaque 3 L rectangular tank that could be divided in half with a Perspex divider. Each end of the tank had distinct visual cues (1.5 cm diameter black spots versus vertical 0.5 cm wide black and white stripes, matched for luminosity). After habituation to the apparatus and handling, we determined the basal preference for each fish: individual fish were placed in the tank for 10 min and the time spent at either end determined using a ceiling mounted camera and Ethovision tracking software (Noldus, Wageningen, NL). Any fish showing >70% preference for either end was excluded from further analysis (between 10% and 20% of fish). Fish were then conditioned with nicotine in the least preferred environment for 20 min, on three consecutive days: Each day each fish was restricted first to its preferred side for 20 min in fish water and then to its least preferred side with nicotine or, if a control fish, vehicle (fish water) added, for another 20 min. After 20 min in the nicotine (or vehicle)-paired environment each fish was returned to its home tank. After 3 days of conditioning, on the following day, fish were subject to a probe trial whereby each fish was placed in the conditioning tank in the absence of divider and the time spent at either end of the tank over a 10 min period was determined as for assessment of basal preference. The change in place preference was determined as the proportion time spent in the nicotine-paired zone during the probe trial minus the proportion time spent in the nicotine-paired zone during basal testing. The CPP procedure has been used and validated previously with nicotine as well as other drugs (*Kily et al., 2008*; *Brock et al., 2017*; *Parker et al., 2016*) .

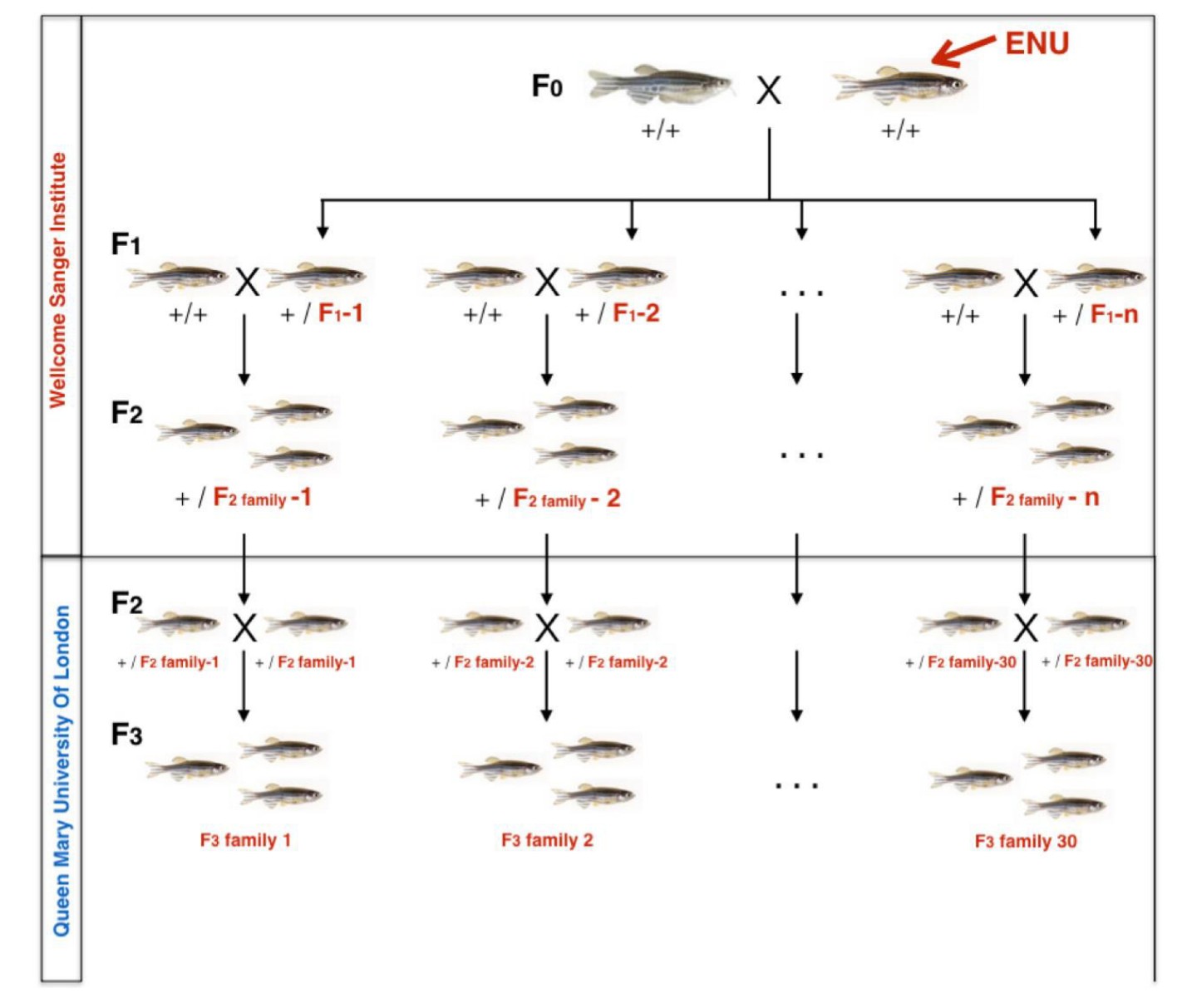

**Figure 10.** Zebrafish breeding scheme to generate F₃ families. F₂ ENU-mutagenized zebrafish, heterozygous for multiple mutations across the entire genome were obtained from the Wellcome Sanger Institute as part of the Zebrafish Mutation Project. At QMUL, heterozygous F₂ fish were incrossed to generate 30 F₃ families, each containing 10–20 nonsense or essential splice site mutations and about 1500 additional exonic and intronic point mutations. F₃ Families were arbitrarily numbered 1–30. Expected mutation rate and type of mutations in coding regions are specified on the right hand side.

Data analysis: Change in preference score was calculated as proportion time spent in drug paired stimulus after conditioning minus proportion time spent in drug paired stimulus before conditioning. Population means between generations were compared using independent two-sample t-tests, and effect-sizes ascertained using Cohen's d (**Cohen, 1992**). For the rescreen of outlier sibling families and *slit3^{sa202}* line, mutant lines were compared with wild type controls using an independent two-sample t-test.

## CPP in the presence or absence of antagonists

To assess the ability of compounds varenicline (Sigma, Gillingham, UK, PZ0004), buproprion (Sigma, Gillingham, UK, B1277) or amisulpride (Tocris, Bristol, UK, C2132) to inhibit subjective effects of

nicotine a modified version of the CPP procedure was used (*Papke et al., 2012*). In this modified version, following habituation and establishment of basal preference, each day each fish was restricted first to its preferred side for 20 min in fish water and then removed from the conditioning tank and transferred to a tank containing the appropriate concentration of test compound or fish water (plus carrier where required) for 10 min. After 10 min the fish was returned to its least pre-ferred side in the conditioning tank with nicotine or, if a control fish, vehicle (fish water) for another 20 min. After 20 min in the nicotine (or vehicle)-paired environment, each fish was returned to its home tank. After three days of conditioning to nicotine in the presence or absence of test com-pound, on the following day, fish were subject to a probe trial whereby each fish was placed in the conditioning tank in the absence of divider and the time spent at either end of the tank over a 10 min period determined. To assess the ability of varenicline (0–20 µM) or buproprion (0–10 µM) to inhibit subjective effects of nicotine in wild type fish, fish were incubated in the presence and absence of increasing doses of test compound (or vehicle) for 10 min before conditioning to 10 µM nicotine. Statistical analysis was performed using a univariate analysis of variance (ANOVA), followed by Tukey's post hoc test.

To test the effect of amisulpride on nicotine-induced CPP in wild type and *slit3*$^{sa1569}$ mutant fish, fish were incubated in the presence or absence of 0.5 mg/L amisulpride for 10 min before condition-ing to 5 µM nicotine. We selected 0.5 mg/L amisulpride as this concentration has been shown to impact behaviour in adults previously (*Tran et al., 2015*) and wild type and homozygous mutants were found to be differentially sensitive to the effect of this concentration on acoustic startle response. Although previous studies with amisulpride in adult zebrafish used a pre-incubation period of 30 min, to be consistent with our analysis of inhibitory effects of varenicline and bupropion, and to reduce the possibility of prolonged confinement in a small volume of drug affecting stress levels which may confound results, we used a preincubation period of 10 min. Two-way ANOVA was per-formed with genotype (*slit3*$^{+/+}$ *and slit3*$^{sa1569/sa1569}$) and treatment (control, nicotine, nicotine+-amisulpride) as independent variables. Values of $p < 0.05$ were considered significant.

## Breeding and selection to assess heritability of nicotine-induced place preference

To test whether nicotine preference is heritable, fish falling in the upper and lower deciles of the 'change in place preference' distribution were kept for analysis and further breeding. Individuals were bred (in-cross of fish from the upper decile and in-cross of fish from the lower decile done sep-arately) and their offspring screened for CPP (Second Generation CPP analysis). The same approach was repeated again: fish at the extremes of the Second Generation CPP distribution curve were selected and in-crossed, and their offspring were used to perform a Third Generation CPP analysis.

## Identification of ENU-induced mutations influencing nicotine place preference

To investigate whether ENU-induced mutations affect fish sensitivity to the rewarding effects of nico-tine, candidate families were selected when the 3–4 fish from a family tested clustered together at one or other extreme of the change in preference distribution curve. To confirm the genetic effect on the CPP phenotype in candidate families, all remaining siblings of that family were assessed for nicotine induced CPP, along with non-mutagenized TLF control fish, to confirm the genetic effect on the phenotype. The experimental team and technical staff were blind to the fish genotype.

Candidate mutations, obtained from exome sequencing on F$_1$ fish, were assessed for co-segrega-tion with behaviour using site specific PCR (*Hamajima et al., 2000*) on genomic DNA. Once a co-segregating candidate mutation was identified, larvae from an independent line carrying a predicted loss of function allele in the same gene were obtained from the Sanger Institute (*slit3*$^{sa202}$) to confirm the association. Heterozygous *slit3*$^{sa202/+}$ and sibling *slit3*$^{+/+}$ larvae were reared to adulthood and assessed for nicotine-induced CPP as described above. All fish were fin clipped and genotyped fol-lowing CPP.

## Zebrafish genomic DNA extraction

Genomic DNA was extracted from fin-clips using QIAGEN DNeasy Blood and Tissue Kit (Qiagen, Manchester, UK) according to manufacturer's instructions. Samples were eluted into distilled water and stored at −20°C until later use.

## Site-specific polymerase chain reaction

Allele-specific PCR SNP assays were used for genotyping $F_3$ individuals for mutations known to be present in the ENU-mutagenized $F_1$ generation. Four primer pairs were designed to carry out PCR genotyping as previously described (*Hamajima et al., 2000*). The list of loss-of-function mutations in the AJBQM1 and AJBQM2 lines is detailed in *Supplementary file 1A*. For each line, a primer was designed with 3' complementary to the ENU-SNP with a second primer ~100 bp downstream. The second pair had one primer with 3' complementary to the wild type base with a second primer ~200 bp upstream. The resulting PCR results in a 300 bp fragment that spans the region and acts as an internal control for the PCR plus one 100 bp fragment if homozygous for the mutation, 2 bands of 100 bp and 200 bp if heterozygous, and one 200 bp fragment if homozygous wild type. The 4-primer groups were designed with melting temperatures as close as possible using the NCBI primer design tool and were ordered from Eurofins, MWG operon (Ebersberg, DE).

## Characterization of larvae

### Antibody staining

In order to visualize axonal pathways, fluorescent immunohistochemistry was carried out in three day old embryos from wild type *slit3$^{+/+}$*, and homozygous mutant *slit3$^{sa1569/sa1569}$* in-crosses. To prevent skin pigmentation, embryos were incubated in 0.2 mM of 1-phenyl 2-thiourea (Sigma, Gillingham, UK) from 24 hr after fertilization. At three days, they were fixed in 4% paraformaldehyde (Sigma, Gillingham, UK) to avoid tissue degradation. For the immunostaining, rabbit polyclonal anti-tyrosine hydroxylase primary antibody (1:200; Sigma, Gillingham, AB152), rabbit polyclonal anti-serotonin (5HT) antibody (1:200, Sigma, Gillingham, S5545) and mouse anti-acetylated tubulin monoclonal antibody (1:1000; Sigma Gillingham, UK, T6793) were used. The three primary antibodies were detected with Alexa 546-conjugated secondary antibodies (1:400; Fisher Scientific, Loughborough, UK A11010). Whole-mount immunohistochemistry and mounting was performed as described previously (*Driever et al., 2012*).

### Confocal microscopy imaging and analysis

Images were acquired using a Leica SP5 confocal microscope. Confocal z-stacks were recorded under the same conditions using diode laser and images were processed under ImageJ environment. Areas of interest for quantification were isolated, making sure that for all the individuals the same number of Z stacks, (covering the same dorsal/ventral distance) were included. The number of cells was quantified using the ImageJ plugin '3D Objects counter' (https://biii.eu/3d-objects-counter). To quantify the number of catecholaminergic axons crossing the midline, a line was drawn from the Medulla oblongata interfascicular zone and vagal area to the locus coeruleus (*Figure 5—figure supplement 1*). Every 10 stacks (~7.6 microns), the number of intensity peaks (defined as grey value intensity >20) was measured. Unpaired t-tests were calculated to assess genotype differences in the number of cells and intensity peaks.

### Startle response in the presence or absence of amisulpride

Five-day-old larvae, generated from adult *slit3* wild type and homozygous mutant (*slit3$^{sa1569/sa1569}$*) fish as for quantitative PCR, were individually placed in 24 well plates. In the drug-free condition, each well contained 300 μL system water and 0.05% of dimethyl sulfoxide (DMSO, Sigma, Gillingham, UK). In the pharmacological conditions, serial dilutions of the dopaminergic and serotonergic antagonist amisulpride (Tocris, Bristol, UK, 71675-86-9) were prepared to give final concentrations of 0.05 mg/L, 0.1 mg/L or 0.5 mg/L amisulpride in 0.05% DMSO. Amisulpride concentrations were chosen based on previous studies in zebrafish (*Tran et al., 2015*) and correspond to 50, 100 and 500 times its Ki value for the D2 receptor in mammals (*Schoemaker et al., 1997*; *Perrault et al., 1997*). To ensure that larvae were exposed to the drug for the same amount of time, amisulpride was added 15 min before undertaking the experiment. Care was taken regarding the distribution of

concentrations and genotypes to ensure that experimental groups were randomly distributed in the plates. Plates were placed in a custom-made filming tower with a tapping device that applied 10 sound/vibration stimuli with two seconds interval between them. The setup for this device has been described elsewhere (Parker and Brennan, 2016). Larval movement was recorded using Ethovision XT software (Noldus Information Technology, Wageningen, NL) and data were outputted in one second time-bins. Three technical replicates were performed (three different days) with three 24 well plates assayed each day.

For each fish, distance travelled (mm) during one second after each tap was recorded. All videos used to evaluate responses were checked for tracking errors. Any points where tracking errors were detected were removed. In total 105 points (31 wild type, 33 heterozygous, 41 homozygous) out of 5460 were removed due to tracking errors. We also excluded individuals that failed to respond to all 10 taps using the criteria defined below. 39 wild type, 38 heterozygous, and 39 homozygous individuals were removed (a total of 115 individuals out of 546).

To evaluate how amisulpride dose and *slit3* genotype affect habituation to the startle stimulus, we defined a response/non-response status for each fish. Threshold for response status was defined as mean distance moved per second before the first stimulus (basal distance) plus two standard deviations (SD) of the mean. Since genotype was not a significant predictor of basal distance travelled – confirmed by a linear mixed effect model with basal distance as the response variable and genotype as the explanatory variable (p=0.123), we calculated the population mean and SD of basal distance for the three *slit3* genotypes together (mean = 1.4 mm, SD = 1.6, mean+2SD = 4.6 mm). Each fish was assigned as 'responder' if it moved more than the threshold in the first second after the stimulus or as 'non-responder' if it did not.

The percentage of fish responding to stimulus together with amisulpride dose, stimulus event number and genotype group were modelled in a beta regression conducted using the R package 'betareg'. The proportion of individuals responding was the response variable and the three-way interaction between amisulpride dose, genotype, and stimulus event number was the explanatory variable. To determine whether this interaction was a significant predictor of individual responsiveness (indicating that genotypes varied in how amisulpride dose affected their habituation to repeated stimuli), likelihood ratio tests for nested regression models were performed. Results of all statistical analyses were reported with respect to a type-1 error rate of $\alpha$ = 0.05.

## Real-time quantitative PCR

Adult *slit3* wild type and *slit3^{sa1569}* homozygous mutant fish, generated from a *slit3^{sa1569/+}* heterozygous in cross, were bred to generate homozygous wild type, heterozygous mutant and homozygous mutant larvae. Embryos were carefully staged at 1, 24 and 48 hr and at five day post fertilisation to ensure, based on morphological criteria, there were no differences in development between groups. mRNA from 3 samples of five-day-old embryos (n = 10 pooled embryos per sample) for each genotype was isolated using the phenol-chloroform method. cDNA was generated using the ProtoScript II First Strand cDNA Synthesis Kit (NEB (UK Ltd.), Hitchen, UK). Relative qPCR assays were performed using the LightCycler 480 qPCR system from Roche Diagnostics, Ltd. with all reactions carried out in triplicates. Reference genes for all the qPCR analyses were *β-actin*, *ef1α* and *rpl13α* based on previous studies (Parker et al., 2016; Tang et al., 2007; Collier and Echevarria, 2013). Accession numbers and primer sequences for the genes can be found in *Supplementary file 1B*.

Relative mRNA expression in qPCR was calculated against reference gene cycle-threshold (Ct) values, and then subjected to one-way ANOVA. To account for multiple testing a Bonferroni correction was applied, and significance was declared at a threshold of 0.001.

## Human cohorts

In London human subjects were recruited from three clinical groups: patients with chronic obstructive pulmonary disease (COPD) (Cohort 1; n = 272); patients with asthma (Cohort 2; n = 293); and residents and carers in sheltered accommodation, with neither condition (Cohort 3; n = 298). The methods used for recruitment and definition of phenotypes are reported elsewhere (Martineau et al., 2015a; Martineau et al., 2015b; Martineau et al., 2015c). The studies were approved by East London and The City Research Ethics Committee 1 (09/H0703/67, 09/H0703/76 and 09/H0703/112). Written informed consent was obtained from all participants.

Details of the Finnish twin cohort are reported elsewhere (*Loukola et al., 2014*; *Loukola et al., 2008*; *Broms et al., 2012*). In brief, twin pairs concordant for moderate to heavy smoking were identified from the population-based Finnish Twin Cohort survey responders. The twin pairs and their siblings were invited to a computer-assisted, telephone-based, structured, psychiatric interview (SSAGA) (*Loukola et al., 2014*), to yield detailed information on smoking behaviour and nicotine dependence as defined by Fagerström Test for Nicotine Dependence (FTND) and DSM-IV diagnoses. Human phenotypes to be investigated in relation to zebrafish nicotine seeking behaviour were determined by consensus a priori.

Sample characteristics of the human cohorts can be found in *Supplementary file 1C*.

## Phenotype definitions for the London cohorts

*Amount smoked* was defined as the average number of cigarettes smoked per day (CPD) for each participant. Participants met criteria for *smoking cessation* if they reported being 'ever smokers' and reported **not** smoking currently. The percentage of current smokers in the cohort was 42%, 7% and 18% for ViDiCO, ViDiAs and ViDiFLU, respectively.

## Phenotype definition for the Finnish twin cohort study

Definitions of the phenotypes were adapted from *Broms et al. (2012)*.

### Amount smoked

Cigarettes per day (CPD) constitutes of eight categories: 1–2, 3–5, 6–10, 11–15, 16–19, 20–25, 26–39, $\geq$40 CPD. In the statistical analyses of the CPD variables, original categorical observations were replaced with class means of CPD (1.5, 3.5, 8, 13, 17.5, 22.5, 32.5, and 45 cigarettes per day, respectively). Regression coefficients can therefore be interpreted as the average change in number of cigarettes smoked per day when the number of minor allele is increased by one.

- CPD: Number of cigarettes smoked per day during month of heaviest smoking. Values ranged from one to >40 with mean = 19.8 cigarettes per day.
- Maximum CPD: Maximum number of cigarettes ever smoked during one day (24 hr period). Values ranged from 2 to 98 with mean = 30 cigarettes per day.

### Smoking initiation

- Age (years) when started to smoke weekly ("How old were you when you first smoked a cigarette at least once a week for at least two months in a row?"). Values ranged from 6 to 54, mean = 17.3 years.
- Sensation felt after smoking the first cigarette or first puffs. Sensation measured as: While smoking your very first cigarettes, did you 1. like the taste or smell of the cigarette, 2. cough, 3. feel dizzy or light-headed, 4. feel more relaxed, 5. get a headache, 6. feel a pleasurable rush or buzz, 7. feel your heart racing, 8. feel nauseated, like vomiting, 9. feel your muscles tremble or become jittery, 10. feel burning in your throat. Sum score of 10 questions (items #1, #4, and #6 were reverse-scored before summation): 0 points if answered 'No', 1 = 'A little bit', 2="Some', 3='Quite a bit', 4="A great deal'. Cronbach's alpha = 0.70. Values ranged from 3.6 to 15.8. Mean = 10.2.

### Nicotine dependence

- DSM-IV ND diagnosis: Nicotine dependence by DSM-IV diagnosis ($\geq$3 symptoms out of 7 occurring within a year). Prevalence = 53.5%.
- DSM-IV ND symptoms: Number of DSM-IV ND symptoms from 0 to 7. Mean = 3
- FTND ($\geq$4): Nicotine dependent if $\geq$4 out of 10 points in Fagerström Test for Nicotine Dependence. Prevalence = 50.4%
- FTND score: Fagerström Test for Nicotine Dependence score: 0 to 10 points. Mean = 3.7.

### FTND time to first cigarette (TTF)

Time to first cigarette in the morning (one item of the FTND scale). Five categories: 0–5 min, 6–15 min, 16–30 min, 31–60 min, >60 min. Categorization differs from original four categories (*Xu et al., 2015*), i.e., 6–30 min is split into 6–15 min and 16–30 min. In our data set 46% of smokers belong to the group of 6–30 min, and from the smoking behaviour point of view there is a significant difference whether one smokes the first cigarette within 6 min or 30 min from waking up. In this data set 22% of smokers belong to the 6–15 min and 24% to the 16–30 min group. Values ranged from 1 to 5 with a mean = 3.1.

### Human genotyping

For the London cohorts, DNA from participants was extracted from whole blood using the salting-out method (*Miller et al., 1988*) and normalized to 5 ng/μL. 10 ng DNA was used as template for 2 μL TaqMan assays (Applied Biosystems, Foster City, CA, USA) performed on the ABI 7900HT platform in 384-well format and analysed with Autocaller software. Pre-developed assays were used to type all SNPs. See *Supplementary file 1* Table 4 for primer and reporter sequences. Typing for two SNP (rs6127118 and rs11574010) failed. For the Finnish cohort, DNA was extracted from whole blood and genotyping was performed at the Wellcome Trust Sanger Institute (Hinxton, UK) on the Human670-QuadCustom Illumina BeadChip (Illumina, Inc, San Diego, CA, USA), as previously described (*Loukola et al., 2014*; *Loukola et al., 2008*; *Broms et al., 2012*). Genotyping and imputation for the Finnish cohort at the Wellcome Trust Sanger Centre have been described previously (*Loukola et al., 2015*).

### Human association analyses

We attempted to replicate the zebrafish findings initially in a cohort from London using a narrow set of SNPs in *SLIT3*. We then used the same set of SNPS to evaluate effects on more detailed smoking phenotypes in a Finnish twin cohort.

London cohort association analysis was performed using PLINK v1.07 (*Purcell et al., 2007*). *SLIT3* SNPs that had been previously associated with disease phenotype were identified and the 20 with low linkage disequilibrium score selected for analysis. Of twenty *SLIT3* SNPs, one departed from Hardy-Weinberg equilibrium (rs13183458) and was excluded. Linear regression was performed on average number of cigarettes smoked per day, controlling for age, sex and cohort. Since physiological and genetic mechanisms may be different in heavy (more dependent) and light (less dependent) smokers we repeated on heavy smokers ($\geq$20 cigarettes per day) and light smokers (<20 cigarettes per day). Smoking cessation (current *vs* ever smokers) was analysed using logistic regression controlling for age, sex and cohort. All analyses were performed under the additive genetic model and multiple testing was taken into account using the Benjamini-Hochberg adjustment. Only individuals from European ancestry were included in analyses.

Association analyses for the Finnish Twin Cohort were performed using GEMMA v0.94 (*Zhou and Stephens, 2012*) with linear mixed model against the allelic dosages controlling for age and sex. Sample relatedness and population stratification were taken into account by using genetic relatedness matrix as random effect of the model.

## Acknowledgements

Funding: MRC UK, G1000403 (CHB/RW); NC3Rs G1000053 (CHB); BBSRC BB/M007863 (CHB); NIH U01 DA044400-03 (CHB, EMB); NIHR PGfAR RP-PG-0609–10181 (RW); NIHR PGfAR, RP-PG-0407–10398 (ARM). CHB is a member of the Royal Society Industry Fellows' College. RW is an NIHR Senior investigator (NF-SI-0515–10076). VK holds a Wellcome Trust Clinical Research Fellowship. JK is supported by the Academy of Finland (grants 308248, 312073). Some of the data in this manuscript have been published previously as a preprint: bioRxiv 453928: DOI: https://doi.org/10.1101/453928.

# Additional information

## Funding

| Funder | Grant reference number | Author |
|---|---|---|
| Wellcome Trust | Clinical research fellowship WT 110284/Z/15/Z | Valerie Kuan |
| National Institutes of Health | Project grant | Caroline Helen Brennan |
| Medical Research Council | G1000403 | Caroline Helen Brennan Robert T Walton |
| National Centre for the Replacement, Refinement and Reduction of Animals in Research | G1000053 | Caroline Helen Brennan |
| Biotechnology and Biological Sciences Research Council | BB/M007863 | Caroline Helen Brennan |
| National Institute for Health Research | PGfAR RP-PG-0609-10181 | Robert T Walton |
| National Institute for Health Research | NIHR PGfAR RP-PG-0407-10398 | Adrian Martineau |
| National Institutes of Health | U01 DA 044400-03 | Caroline Helen Brennan Elisabeth M Busch-Nentwich |
| National Institute for Health Research | NF-SI-0515-10076 | Robert T Walton |
| Royal Society | Industry Fellows College | Caroline H Brennan |
| Academy of Finland | 308248 | Jaakko Kaprio |
| Academy of Finland | 312073 | Jaakko Kaprio |

The funders had no role in study design, data collection and interpretation, or the decision to submit the work for publication.

## Author contributions

Judit García-González, Formal analysis, Investigation, Visualization, Methodology, Writing - original draft, Writing - review and editing; Alistair J Brock, Formal analysis, Investigation, Visualization; Matthew O Parker, Formal analysis, Investigation; Riva J Riley, Teemu Palviainen, Valerie Kuan, Formal analysis; David Joliffe, Ari Sudwarts, Muy-Teck Teh, Investigation; Elisabeth M Busch-Nentwich, Derek L Stemple, Resources; Adrian R Martineau, Jaakko Kaprio, Resources, Supervision, Methodology; Robert T Walton, Supervision, Funding acquisition, Methodology, Project administration; Caroline H Brennan, Conceptualization, Formal analysis, Supervision, Funding acquisition, Validation, Investigation, Visualization, Methodology, Project administration

## Author ORCIDs

Judit García-González (ID) https://orcid.org/0000-0001-6245-740X
Matthew O Parker (ID) https://orcid.org/0000-0002-7172-5231
Riva J Riley (ID) https://orcid.org/0000-0001-5708-7424
Muy-Teck Teh (ID) https://orcid.org/0000-0002-7725-8355
Elisabeth M Busch-Nentwich (ID) https://orcid.org/0000-0001-6450-744X
Derek L Stemple (ID) https://orcid.org/0000-0002-8296-9928
Adrian R Martineau (ID) https://orcid.org/0000-0001-5387-1721
Jaakko Kaprio (ID) https://orcid.org/0000-0002-3716-2455
Teemu Palviainen (ID) https://orcid.org/0000-0002-7847-8384
Valerie Kuan (ID) https://orcid.org/0000-0001-7873-6972
Robert T Walton (ID) https://orcid.org/0000-0001-7700-1907
Caroline H Brennan (ID) https://orcid.org/0000-0002-4169-4083

## Ethics

Human subjects: The London studies were approved by East London and The City Research Ethics Committee 1 (09/H0703/67, 09/H0703/76 and 09/H0703/112) Written informed consent was obtained. The Finnish cohort was ascertained from the Finnish Twin Cohort study. Data collection took place in 2001-2005. The study was approved by the Ethics Committee of the Hospital District of Helsinki and Uusimaa, Finland. All participants provided written informed consent.

Animal experimentation: This study was performed in strict accordance with the UK Animals (Scientific Procedures) Act 1986. All of the animals were handled according to approved Home Office protocols (PPL P6D11FBCD) and following approval from the institutional animal welfare and ethics review board (AWERB) of Queen Mary University of London.

## Decision letter and Author response

Decision letter https://doi.org/10.7554/eLife.51295.sa1
Author response https://doi.org/10.7554/eLife.51295.sa2

# Additional files

## Supplementary files

• Supplementary file 1. Supplementary methods and results for zebrafish genotyping and gene expression analysis, and human association analyses. Supplementary methods: (**A**) Loss of function mutations present in AJBQM1 and AJBQM2 founders; (**B**) Gene identifiers and primer sequences used for zebrafish gene expression analysis; (**C**) Human association study sample characteristics; (**D**) Primer and reporter sequences used for human genotyping. Supplementary results: (**H**) Results of site specific PCR genotyping of AJBQM1 and AJBQM2; (**F**) Logistic regression analysis for human association analysis on smoking initiation; (**G**) Logistic regression analysis for human association analysis on persistent smoking; (**H**) Full qPCR gene expression results for *slit3* wildtype and *slit3*$^{sa1569/sa1569}$ homozygous mutants.

• Transparent reporting form

## Data availability

All zebrafish data generated or analysed during this study are included in the manuscript and supporting files. Source data files have been provided for Figures 1 - 8. Human genomic data. Information for the London cohorts can be found in https://clinicaltrials.gov - ViDiCO:ID NCT00977873 - ViDiAs: ID NCT00978315 - ViDiFLU : NCT01069874. Controlled access to the data is managed by The Pragmatic Clinical Trials Unit (PCTU), QMUL (https://www.qmul.ac.uk/pctu/collaborate-with-us/data-sharing/. Data may be shared either by transferring the data out of the PCTU secure servers or by granting the recipient researcher access to the data while it remains on the PCTU secure servers. A Data Sharing Agreement is required for the former, and a Data Access Request for the latter. A partial dataset, including baseline characteristics and outcomes, stripped of all identifying information, can also be obtained from a.martineau@qmul.ac.uk. For the Finnish cohorts: Genotype and phenotype Finnish twin datasets are deposited in the Biobank of the National Institute for Health and Welfare (https://thl.fi/en/web/thl-biobank/for-researchers). Qualified applicants (academics or companies) can apply for the data (genotypes and phenotypes) from the Biobank via this web site (https://thl.fi/en/web/thl-biobank/for-researchers). Applications are reviewed and granted by the Director of the Biobank on a case by case basis. No analytic subsets are identified externally, but where a request for access is accepted, the Biobank will identify the data linked to this publication.

The following previously published dataset was used:

| Author(s) | Year | Dataset title | Dataset URL | Database and Identifier |
|-----------|------|---------------|-------------|-------------------------|
| Loukola A, Broms U, MaunuH, Widén E, Heikkilä K, Siivola M, Salo A, Pergadia ML, Ny- | 2014 | Finnish Twin Cohort | https://wiki.helsinki.fi/display/twineng/Twinstudy | Nicotine Addictions Genetics Study - Finnish site, Twinstudy |

man E, Sammalisto
S, Perola M, Agra-
wal A, Heath AC,
Martin NG, Mad-
den PA, Peltonen L,
Kaprio J

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
