## [Decision Letter]

**Acceptance summary:**

This manuscript uses an animal model, the zebrafish, to identify mutations that increase nicotine preference. The authors then proceed to identify variants of the same gene in humans and report correlations with smoking. The combined observations in two so distantly related species adds strength to the findings.

**Decision letter after peer review:**

Thank you for submitting your article "Identification of *Slit3* as a locus affecting nicotine preference in zebrafish and human smoking behaviour" for consideration by *eLife*. Your article has been reviewed by two peer reviewers, and the evaluation has been overseen by a Reviewing Editor and Didier Stainier as the Senior Editor. The following individuals involved in review of your submission have agreed to reveal their identity: Lars Westberg (Reviewer #1); Karl Clark (Reviewer #2).

The reviewers have discussed the reviews with one another and the Reviewing Editor has drafted this decision to help you prepare a revised submission.

Summary

Genetic contributions to smoking behaviour were sought by screening mutagenized zebrafish for nicotine place preference. A SNP in the *Slit3* gene was identified among 300 predicted loss of function mutants. Two human cohorts with SNP variants in the SLIT3 gene showed correlation with cigarette consumption and likelihood of cessation. Zebrafish larvae with *Slit3* mutations displayed altered behavioural sensitivity to amisulpride, an antagonist at dopamine and serotonin receptors. Larvae had increased mRNA of receptor 5htr1aaNo effect on neuronal. These findings show a role for SLIT3 signalling in development of pathways involved in nicotine effects.

Essential revisions:

The reviewers and myself as reviewing editor find this to be an interesting study but we request more details to evaluate the robustness of the findings as detailed below. We also request more discussion of the suggested role of monoamines for slitr3 actions on nicotine preference and improved description of the human genetic study.

1) Please provide more quantitative parameters in the Abstract and also state clearly in what direction the nicotine preference was affected: Out of 25 identified inactivating mutations, one in the *Slit3* gene segregated with increased nicotine preference in heterozygous individuals.

2) Explain in the Abstract why amisulpride was tested: "...dopaminergic and serotonergic antagonist known to affect startle reflex which is correlated with addiction..."

3) Say in what direction amisulpride affected the behavior.

4) Has amisulpride been demonstrated to have the same receptor preference and effect (antagonist) in zebrafish as in human? Amisulpride was not used in the zebrafish study in reference 46. Discussion paragraph five says that "acoustic startle is sensitive to modulation of dopaminergic and serotonergic signalling in all species studied", but as the three references provided concern only three species (human, mouse and zebrafish), it is better to say "in the three species studied".

5) The results from the mutant screen showing an association between slitr3 and nicotine preference in zebrafish seems convincing, although it is based on a moderate number of fish. Please consider presenting this result as a main finding in the Abstract and Discussion, rather than the tentative associations from the human genetic study.

6) The results from the pharmacological and qPCR experiments are interesting but deserves a more thorough discussion on how the findings using the D2 antagonist amisulpride, fits together with the difference in htr1aa expression and lack of difference in TH expression.

7) The benefits and the limitations of using zebrafish of various ages in the different experiments (behavioral, IHC, qPCR vs. pharmacology) should be more clearly addressed.

8) The method used for quantification of the immunostainings should be described. It is not possible to exclude group differences from the pictures in presented in Figure 5A-C.

9) Whole embryos were used for the qPCR study. This should be raised as a limitation, since many of the studied genes most likely are not brain-specific.

10) Motivate the chosen genes investigated with qPCR. For example, some of the crucial enzymes were not studied.

11) You may consider to compare serotonin neurons using the immunohistochemical approach they used for TH neurons in Figure 5.

12) Explain why different scales are used in Supplementary file 1–table 5A and B.

13) The description of the human genetic study needs to be improved. For example, the number of participants included from both of the genetic cohorts should be presented in Materials and methods (but also in Results, e.g. in Tables 1 and 2).

Furthermore, it is unclear if the Finish samples were genotyped by the authors or if data was received from somewhere. Also consider modifying the study design. It is not clear why the London sample was divided into heavy and light smokers. Already before splitting the sample the number of participants must be considered as low for a genetic study. With the current study design, very many tests were conducted, so the correction method for multiple testing correction must be presented (like Bonferroni for qPCR data). You may consider to use the London study as an exploratory cohort and the Finish study as a replication cohort. In the Finish cohort only SNPs significantly associated in the exploratory cohort would be investigated.

14) Explore to what extent SLIT3 gene variants have been associated with nicotine dependence and related phenotypes in previous GWA studies. One tool that may be helpful is the GWAS ATLAS (https://atlas.ctglab.nl).

15) It would be helpful if a picture was added for the gene with the SNP locations indicated, above the LD plot, in Figure 9.

16) Although Figure 5 is meant to show there are not gross abnormalities in axon pathfinding, I feel that there are observable differences between the WT and (*Slit3*^+/-^ OR *Slit3*^-/-^) in acetylated tubulin staining. I am not sure if it is the particular panels chosen or a real but subtle difference. If there are similar individual differences between any 2 WT, it may be better to show 2 individuals for each genotype rather than the partial Z-stack that is in panel B. If it is a real, but subtle difference, it is better to discuss than try to pass off as the same.

17) The habituation data were found to be unconvincing. Most habituation in zebrafish larvae is a yes or no startle response and not based on distance travelled. However, these assays are typically earlier in development, so this may be a bit of an adaptation and could account for some of the differences. That being said, in Panel A the mean response based on your cutoff of 2.5 mm of movement does not reach habituation in your trial. This may lead to the less than spectacular differences in habituation rates. You probably want to be at a position where 80% of wild-type fish are showing habituation. This could be achieved by using more than 10 stimuli, decreasing the intensity of the stimuli, or changing the distance moved criteria. In other words, you may be too close to the arbitrary cutoff of 50% to be measuring real changes in habituation. So currently, the interpretation of these results may be a bit of a stretch. Even though statistical differences were measured, they may not be biologically relevant.

18) What were the overall results of this screen of 30 families? It is suggested that there were families with reduced nicotine CPP and increased nicotine CPP. Two mutations were described, but out of how many negative and positive effector families?

19) In Figure 3, the AJBQM2 line does not show significance in comparison to WT nor WT (nic) based on letter superscript (a), yet it is discussed as it does, including line "AJBQM2 differed from wildtype nicotine exposed fish but not wildtype saline controls." If it is not significantly different there are several places in the manuscript that are misleading.

---

## [Author Response]

Essential revisions:The reviewers and myself as reviewing editor find this to be an interesting study but we request more details to evaluate the robustness of the findings as detailed below. We also request more discussion of the suggested role of monoamines for slitr3 actions on nicotine preference and improved description of the human genetic study.1) Please provide more quantitative parameters in the Abstract and also state clearly in what direction the nicotine preference was affected: Out of 25 identified inactivating mutations, one in the Slit3 gene segregated with increased nicotine preference in heterozygous individuals.

We have now edited the Abstract including:

– Number of mutagenized families screened.

– Further details on the identification of Slit3 and direction of the effect: "Out of 25 identified loss-of-function mutations, one in the *slit3* gene segregated with increased nicotine preference in heterozygous individuals."

– Sample size for the human studies "Focussed SNP analysis of the homologous human locus in cohorts from UK (n=863) and Finland (n=1715)"

2) Explain in the Abstract why amisulpride was tested: "....dopaminergic and serotonergic antagonist known to affect startle reflex which is correlated with addiction..."

This sentence has been added now in the Abstract: "Characterisation of *slit3* mutant larvae and adult fish revealed decreased sensitivity to the dopaminergic and serotonergic antagonist amisulpride, known to affect startle reflex that is correlated with addiction in humans, and increased htr1aa mRNA expression in mutant larvae."

3) Say in what direction amisulpride affected the behavior.

We have replaced in the Abstract the expression "altered behavioural sensitivity to amisulpride" by "decreased sensitivity to.... amisulpride". The direction amisulpride affected the response has also been included in the Results section "Amisulpride caused a biphasic dose dependent effect on stimulus response in wildtypes such that 0.05mg/L caused an increase in responders across all 10 stimuli, and 0.5mg/L caused a decrease (Effect of amisulpride dose p<0.001). A similar pattern was observed for heterozygous *Slit3^sa1569^*, but the effect of amisulpride was not significant (p=0.083). Amisulpride dose had no significant effect on stimulus response in homozygous homozygous *Slit3^sa1569^*, that showed an increase in response to low doses but were less sensitive to inhibition at high doses (Figure 6D-F)."

4) Has amisulpride been demonstrated to have the same receptor preference and effect (antagonist) in zebrafish as in human? Amisulpride was not used in the zebrafish study in reference 46. Discussion paragraph five says that "acoustic startle is sensitive to modulation of dopaminergic and serotonergic signalling in all species studied", but as the three references provided concern only three species (human, mouse and zebrafish), it is better to say "in the three species studied".

We thank the reviewers for pointing out this error. We have now removed the reference 46. We have also modified extensively the discussion and the sentence "acoustic startle is sensitive to modulation of dopaminergic and serotonergic signalling in all species studied" has been replaced. See Discussion, paragraph seven on amisulpride effects.

5) The results from the mutant screen showing an association between slitr3 and nicotine preference in zebrafish seems convincing, although it is based on a moderate number of fish. Please consider presenting this result as a main finding in the Abstract and Discussion, rather than the tentative associations from the human genetic study.

We have now expanded the mutant screen description and results in the Abstract "Of 30 ENU mutagenized families screened, two showed increased or decreased nicotine preference. Out of 25 inactivating mutations in the families, one in the *slit3* gene segregated with increased nicotine preference in heterozygous individuals."

We have also emphasised the mutant screen results in the Discussion: "The aim of this study was to use forward genetic screening in zebrafish to identify loci affecting human smoking behaviour. Among 30 mutant families screened for CPP, we found a family whose members had increased nicotine preference compared to wildtypes. Out of the 13 preidentified loss-of-function mutations in the family, only *slit3* co-segregated with the behaviour. We replicated the increase in nicotine preference using an independent zebrafish mutant line with a loss of function mutation in *slit3*. Next, we established the relevance in humans by identifying two markers in SLIT3 where the presence of the minor allele was associated with fewer cigarettes smoked each day and with smoking cessation"

6) The results from the pharmacological and qPCR experiments are interesting but deserves a more thorough discussion on how the findings using the D2 antagonist amisulpride, fits together with the difference in htr1aa expression and lack of difference in TH expression.

We have extensively re-written the Discussion adding discussion of interactions between htr1a and dopamine, paragraphs eleven and twelve.

7) The benefits and the limitations of using zebrafish of various ages in the different experiments (behavioral, IHC, qPCR vs. pharmacology) should be more clearly addressed.

We have now acknowledged the main benefit and limitations of using zebrafish of various ages in the Discussion: "There are limitations to our findings: we used zebrafish of various ages in different experiments. While this confirms that loss-of-function in *slit3* alters behaviour from early life to adulthood, suggesting a developmental role, mechanisms underlying the two behavioural phenotypes may differ"

8) The method used for quantification of the immunostainings should be described. It is not possible to exclude group differences from the pictures in presented in Figure 5A-C.

In response to this and comment 10 we have included anti-5HT staining and quantification data for both 5HT and TH staining. (subsection "Characterisation of *Slit3^sa1569^* mutants", Figure 5, and Materials and methods subsection "Confocal microscopy imaging and analysis").

9) Whole embryos were used for the qPCR study. This should be raised as a limitation, since many of the studied genes most likely are not brain-specific.

We have included this limitation in the Discussion: "We used whole embryos for the qPCR study so changes in expression in non-neuronal tissue may contribute to the observed differences, further, expression of genes in one tissues may mask changes of expression in another."

10) Motivate the chosen genes investigated with qPCR. For example, some of the crucial enzymes were not studied.

We selected genes for qPCR based on their links in previous studies to addiction and smoking behavior. We have now explained the rationale for the genes selected in the Results section, and added references for links between these genes and smoking.

We acknowledge that some important genes may have been missed by adding the following sentence in the Discussion: "In addition, we only examined a limited number of receptors and transporters for key neurotransmitter pathways. Important differences in other transmitter pathways and neurotransmitter metabolism may have been missed."

11) You may consider to compare serotonin neurons using the immunohistochemical approach they used for TH neurons in Figure 5.

Thank you for the suggestion. We have now added results for immunohistochemical assays labelling serotonergic neurons (Figure 5 and Figure 5—figure supplement 1). As previously described in comment #7, the number of serotonergic cells in wildtype vs homozygous mutant larvae was compared across different brain regions, with no significant differences observed.

12) Explain why different scales are used in Supplementary file 1–table 5A and B.

Supplementary file 1–table 5A and 5B represent the genotyping of zebrafish screened for CPP. Since each fish had a different value for the CPP change score, values in Table 5A (corresponding to the AJQM1 family, where all fish had increased CPP change scores) are greater than values in Table 5B (corresponding to the AJQM2 family, where all fish had decreased CPP change scores).

We have changed the table layout, so each row represents one fish, and columns represent each fish CPP change score, and genotype for each locus assessed (13 loci for AJBQM1, 12 loci for AJBQM2). We have also added a fish ID number to clarify that each row is one fish, and have added the following text to the table legend: "Each row represents one fish, with their corresponding CPP change score, and genotype for each locus assessed (13 loci for AJBQM1, 12 loci for AJBQM2)".

13) The description of the human genetic study needs to be improved. For example, the number of participants included from both of the genetic cohorts should be presented in Materials and methods (but also in Results, e.g. in Tables 1 and 2).

Numbers have now been added to the legends of Tables 1 and 2.

Furthermore, it is unclear if the Finish samples were genotyped by the authors or if data was received from somewhere.

Details of the genotyping are provided in one additional reference which has now been inserted (Loukola et al., 2015).

Also consider modifying the study design. It is not clear why the London sample was divided into heavy and light smokers.

The London sample was divided as genetic mechanisms may vary according to level of consumption – now clarified in subsection "Human association analyses".

Already before splitting the sample the number of participants must be considered as low for a genetic study. With the current study design, very many tests were conducted, so the correction method for multiple testing correction must be presented (like Bonferroni for qPCR data).

We used the Benjamini Hochberg procedure to adjust for multiple comparisons subsection "Human association analyses".

You may consider to use the London study as an exploratory cohort and the Finish study as a replication cohort. In the Finish cohort only SNPs significantly associated in the exploratory cohort would be investigated.

This is an important point and we agree that the sample size is low for a discovery study, but the human samples are being used for replication of the findings in the zebrafish and included only narrow sets of SNPs at a single gene.

We analysed the London sample first, and then followed-up the findings in the Finnish sample. By looking at the same set of SNPs in both samples, we can see the consistency of findings (effect sizes). If we looked in the Finnish sample only at the SNPs significant in the London sample, we would potentially be missing something due to type II errors. Also, the Finnish data had richer and more informative phenotypes.

The rationale has been clarified in subsection "Human association analyses" and paragraph fourteen of the Discussion.

14) Explore to what extent SLIT3 gene variants have been associated with nicotine dependence and related phenotypes in previous GWA studies. One tool that may be helpful is the GWAS ATLAS (https://atlas.ctglab.nl).

Thank you that is very helpful – references to previous associations have been added.

15) It would be helpful if a picture was added for the gene with the SNP locations indicated, above the LD plot, in Figure 9.

A picture for the gene with SNP locations and functional domains has been added.

16) Although Figure 5 is meant to show there are not gross abnormalities in axon pathfinding, I feel that there are observable differences between the WT and (Slit3+/- OR Slit3-/-) in acetylated tubulin staining. I am not sure if it is the particular panels chosen or a real but subtle difference. If there are similar individual differences between any 2 WT, it may be better to show 2 individuals for each genotype rather than the partial Z-stack that is in panel B. If it is a real, but subtle difference, it is better to discuss than try to pass off as the same.

In response to comment 7 and 10 we have changed the figure adding 5HT staining and quantification of 5HT and TH cells and axons. The tubulin figures have been replaced and removed to the Figure 5—figure supplement 1.

17) The habituation data were found to be unconvincing. Most habituation in zebrafish larvae is a yes or no startle response and not based on distance travelled. However, these assays are typically earlier in development, so this may be a bit of an adaptation and could account for some of the differences. That being said, in Panel A the mean response based on your cutoff of 2.5 mm of movement does not reach habituation in your trial. This may lead to the less than spectacular differences in habituation rates. You probably want to be at a position where 80% of wild-type fish are showing habituation. This could be achieved by using more than 10 stimuli, decreasing the intensity of the stimuli, or changing the distance moved criteria. In other words, you may be too close to the arbitrary cutoff of 50% to be measuring real changes in habituation. So currently, the interpretation of these results may be a bit of a stretch. Even though statistical differences were measured, they may not be biologically relevant.

We take the reviewers point and have re-analysed the data using a more stringent response criterion of mean baseline distance travelled per second plus 2 SD. Using this criterion 68% of wildtype individuals respond to the first stimulus, 82% to the first and/or second and 16% to the last. i.e 86% show habituation across the 10 stimuli. The change in criterion alters the p values but the main findings remain the same. We have adjusted the Materials and methods, Results and Discussion sections to reflect the changes.

18) What were the overall results of this screen of 30 families? It is suggested that there were families with reduced nicotine CPP and increased nicotine CPP. Two mutations were described, but out of how many negative and positive effector families?

Out of 30 families screened, individuals from nine different families were in the top 5% of the change in preference distribution, and individuals from five different families in the bottom 5%. We only followed-up the AJBQM1 and AJBQM2 families because they were the only ones where all the members clustered in the top and bottom of the change in preference distribution. This is more clearly stated now in subsection "Identification of *slit3* mutations affecting nicotine place preference in zebrafish".

19) In Figure 3, the AJBQM2 line does not show significance in comparison to WT nor WT (nic) based on letter superscript (a), yet it is discussed as it does, including line "AJBQM2 differed from wildtype nicotine exposed fish but not wildtype saline controls." If it is not significantly different there are several places in the manuscript that are misleading.

The letter superscript (a) was on the AJBQM2 bar by mistake. We thank the reviewers for pointing out this error. We have now edited superscript letters and clarified the meaning of superscript letters in the legend.